Reduction in sunlit Earth reflected radiance 317 to 780 nm during the eclipse of 21 August 2017
Jay Herman[1], Guoyong Wen[2], Alexander Marshak[3], Karin Blank[3], Liang Huang[4], Alexander Cede[5], Nader
Abuhassan[1], Matthew Kowalewski[6]
Abstract

5        Ten wavelength channels of calibrated radiance image data from the sunlit Earth are obtained

every 65 minutes during Northern Hemisphere summer from the DSCOVR/EPIC instrument located near
the Earth-Sun Lagrange-1 point ($L_1$), 1.5 million km from the Earth. The $L_1$ location permitted seven
observations of the Moon's shadow on the Earth for about 3 hours during the 21 August 2017 eclipse.
Two of the observations were timed to coincide with totality over Casper, Wyoming and Columbia,
Missouri. Since the solar irradiances within 5 channels ($\lambda_i$ = 388, 443, 551, 680, and 780 nm) are not
strongly absorbed in the atmosphere, they can be used for characterizing eclipse reduction in reflected
radiances for the sunlit face of the Earth containing the eclipse shadow. Five channels ($\lambda_i$ = 317.5, 325,
340, 688, and 764 nm) that are partially absorbed in the atmosphere give consistent reductions
compared to the non-absorbed channels. This indicates that cloud reflectivities dominate the 317.5 to
780 nm radiances reflected back to space from the sunlit Earth's disk with a strong contribution from
Rayleigh scattering for the shorter wavelengths. A reduction of 10 % in estimated spectrally integrated
radiance (387 to 781 nm) reflected from the sunlit Earth towards $L_1$ was obtained for two sets of
observations on 21 August 2017, while the shadow was in the vicinity of Casper, Wyoming (42.8666° N,
106.3131° W, centered on 17:44:50 UTC) and Columbia, Missouri (38.9517° N, 92.3341° W, centered on
18:14:50 UTC). In contrast, when non-eclipse days (20 Aug. and 23 Aug.) are compared for each
wavelength channel, the change in reflected light is much smaller (less than 1 % for 443 nm compared to
9 % (Casper) and 8 % (Columbia) during the eclipse). Also measured was the ratio $R_{EN}(\lambda_i)$ of reflected
radiance on adjacent non-eclipse days divided by radiances centered in the eclipse totality region with
the same geometry for all 10 wavelength channels. The measured $R_{EN}$(443 nm) was smaller for Columbia
(169) than for Casper (935), because Columbia had more cloud cover than Casper. $R_{EN}(\lambda_i)$ forms a useful
test of 3-D radiative transfer models for an eclipse in the presence of optically thin clouds. Specific
values measured at Casper with thin clouds are $R_{EN}$(340 nm) = 475, $R_{EN}$(388 nm) = 3500, $R_{EN}$(443 nm) =
935, $R_{EN}$(551 nm) = 5455, $R_{EN}$(680 nm) = 220, and $R_{EN}$(780 nm) = 395. Some of the variability is caused by
changing cloud amounts within the moving region of totality during the 2.7 minutes needed to measure
all 10 wavelength channels.
Keywords: Atmospheric Processes, Eclipse, DSCOVR/EPIC, Reflected Energy
**Correspondence email:  jay.r.herman@nasa.gov**
**[1]University of Maryland Baltimore County JCET**
**[2]Morgan State University, Baltimore Maryland**
**[3]NASA Goddard Space Flight Center, Greenbelt, Maryland**
**[4]Science Systems and Applications, Lanham, Maryland**
**[5]Goddard Earth Sciences Technology & Research (GESTAR) Columbia, Columbia, MD 21046, USA**
**[6]SciGlob Instruments and Services, Elkridge, Maryland USA**

## 1.0 Introduction

Measured backscattered radiances of the entire sunlit Earth were obtained during the 21 August 2017 eclipse from EPIC (Earth Polychromatic Imaging Camera) on the DSCOVR (Deep Space Climate Observatory) satellite. EPIC obtains synoptic observations of the Earth from an orbit around the $L_1$ point (Lagrange 1) 1.5 million km from Earth (Herman et al., 2018). EPIC top of the atmosphere TOA albedo measurements, made at a backscatter angle of $172^O$, are in the enhanced reflectivity regime (hotspot angles). EPIC non-eclipse day TOA albedos are compared to the satellite instrument POLDER (POLarization and Directionality of the Earth's Reflectances) surface reflectivity measurements at $8^O$ (Maignan et al., 2004). This study focuses on data from two selected locations during the 21 August 2017 eclipse that crossed the United States from west to east. The locations selected were Casper, Wyoming and Columbia, Missouri, both near the center of the path of totality and both with a nearly overhead total solar eclipse (local time 11:45 in Casper, Wyoming and 13:12 in Columbia, Missouri). The sites were selected in advance to have a high probability of almost cloud-free skies, and so that totality would occur about 30 minutes apart in UTC (Coordinated Universal Time) to accommodate the satellite's ability to acquire data. On the day of the eclipse, Casper, Wyoming had almost clear skies (Fig. 1), with a small amount of thin clouds visible, while Columbia, Missouri had more low altitude cloud cover (Fig. 2).

Observations of total solar eclipses have been made with varying degrees of sophistication for thousands of years as reviewed by Littman et al. (2008). At a given location, observations of reduced irradiance reaching the Earth's surface are limited to just a few minutes of totality and about two hours of partial obscuration (Meeus, 2003). The totality region (umbra) is an oval of about 110 -120 km in size near local noon at Casper, Wyoming and Columbia, Missouri, but will change size and shape as a function of local solar zenith angle (https://eclipse2017.nasa.gov/eclipse-maps). Some of the complicating factors concerning quantitative eclipse observations include the effects of the solar corona and light scattered in the atmosphere (Liendo, and Chacin, 2004; Emde and Mayer, 2007).

A detailed analysis of an eclipse that occurred in 2006 over southern Europe includes both ground-based and space-based polar orbiting MODIS (Moderate Resolution Imaging Spectroradiometer) observations of cloud cover before totality (Gerasopoulos et al., 2008) as well as theoretical modelling of the eclipse, but unlike the present study, it was largely limited to local effects near the region of totality. A comparison from a meteorological radiation model and measurements of total solar irradiance were made near Athens Greece (84 % of a total eclipse) that showed good agreement in the presence of light clouds (Psiloglou and Kambezidis, 2007). A 3D Monte Carlo radiative transfer study (Emde and Mayer, 2007) was applied to the geometry for the nearly overhead total eclipse of 29 March 2006 (13:20 local time in Turkey) to estimate the downward global radiation at the surface, but without the effect of clouds included in the calculation. An application of the 3D model to the 2006 eclipse over Kastelorizo, Greece with fairly cloud-free measurements (few cumulus, 1-2 octas, and scattered cirrus, 3-4 octas) at 380 nm showed good agreement for the ratio (ratio = 217) of global surface irradiance starting 5 minutes before totality to that during totality (Kazantzidis et al., 2007). Successful modelling of the light levels during an eclipse under realistic conditions is the first step toward improved modelling of high cloud reflection and shadowing of solar radiation on the Earth's energy balance.

The observations from the DSCOVR satellite are part of a larger project that combines
simultaneously obtained satellite and ground-based measurements using a pyranometer (Ji and Tsay,
2000) and the Pandora Spectrometer Instrument (Herman et al., 2009) at both sites. The combination
will be used to help validate three dimensional (3D) radiative transfer models applicable to analysis of
eclipse effects on radiances reflected back to space and reaching the Earth's surface. This study presents
the only calibrated spectral synoptic satellite data of the sunlit Earth ever obtained during an eclipse,
which should place tighter limits on validating radiative transfer studies under realistic conditions. The
data includes EPIC measured ozone absorption (316±5 DU Casper and 305±5 DU for Columbia, see Fig.
A3), $O_2$ A- and B-band absorption, clouds, aerosols, and scene and surface reflectivity (Herman et al.,
2018; Marshak et al., 2018).
DSCOVR/EPIC observations of the entire sunlit Earth from the eclipse day, 21 August 2017, are
compared to those from two non-eclipse days to quantify the change of the global integral of reflected
solar radiation caused by the eclipse. We present a potential validation test data set for the 21 August
2017 eclipse for 3D radiative transfer models, namely the ratio of radiances without the eclipse on 20
and 23 August to the same regions that contained totality on 21 August 2017 (based on a suggestion in
the paper by Emde and Mayer, 2007).
Section 2 describes the DSCOVR/EPIC instrument, available data, and monochromatic images based
on measured counts per second, C/s. Section 3.1 presents a comparison between eclipse and non-
eclipse days. Section 3.2 gives an estimate of the global reduction of reflected sunlight during the eclipse
over Casper, WY and Columbia, MO.

**2.0 EPIC Instrument and Data Description**

The EPIC instrument onboard the DSCOVR spacecraft, in a six-month orbit near the $L_1$ point
since June 2015, observed the Moon's shadow for about 3 hours. The EPIC data comprises a set of seven
observations (16:44 to 19:44 UTC) starting in the Pacific Ocean and ending in the Atlantic Ocean, while
synoptically observing the entire sunlit disk of the Earth (nominal size $0.5^O$). EPIC is a 10 wavelength
filter camera with a 2048x2048 pixel CCD (charge couple detector) using a 30-cm aperture Cassegrain
telescope with a field of view (FOV) of $0.62^O$ that continuously points at the sunlit Earth. The sampling
size on the Earth is nominally 8 km at the center of the image with an effective spatial resolution of
10x10 $km^2$ for the 443 nm channel and 17x17 $km^2$ for the other 9 filter channels.  Operation of EPIC
consists of sequentially selecting a filter from 2 rotatable filter wheels and an exposure time using a
rotating disk shutter mechanism. Invariant exposure times were set at the beginning of the on-orbit
mission to fill the CCD wells to about 80 % and avoid blooming (a saturated pixel affecting its neighbors).
The CCD was calibrated for the sensitivity differences between the pixels (flatfielding), and
measurements were made in the laboratory and in-flight to obtain corrections for stray light effects.
Corrections for dark current are applied based on periodic measurements with the shutter closed. EPIC
is kept centered on the Earth during its 6-month north-south tilted Lissajous orbit about the Earth-Sun $L_1$
point. The spacecraft is never closer than $4^O$ from the Earth-Sun line, which makes it possible to observe
an eclipse without the Moon being in the FOV. On 21 August 2017, DSCOVR was 7.7$^O$ from the Earth-Sun
line. A more detailed description of EPIC is given in Herman et al. (2018) and Marshak et al., (2018).

The geolocated EPIC data (Counts per second, C/s) from each set of 10 wavelengths are
contained in an HDF5 formatted file available from the permanent NASA Langley data repository center
(https://eosweb.larc.nasa.gov/project/dscovr/dscovr_epic_l1b). Contained in each Level-2 data HDF5 file
are the 2048 x 2048 array of C/s measured by EPIC and a common latitude and longitude grid. The
geolocated data are organized corresponding to the rectangular CCD grid, 1 data point per CCD pixel.
For the time of the eclipse, the illuminated CCD pixels are within a circular boundary corresponding to
Np = 2.59x10$^6$ illuminated pixels (illuminated pixels formed a circle of 1816 pixels in diameter out of a
maximum of 2048 pixels. To reduce the volume of telemetry data, all measurements, except those from
the 443 nm channel, were 2x2 averaged onboard DSCOVR to 1024 x 1024 pixels. After geolocation onto
a common latitude x longitude grid, the data from all channels are presented as 2048 x 2048 points with
off-earth points represented as the floating point symbol for "infinity". All of the data products (e.g.,
ozone amounts) are also freely available at the above repository center.

The EPIC HDF5 file names (e.g., epic_1b_20170821174450_02.h5) from the NASA data
repository are interpreted as Year 2017, Month 08, Day 21, UTC 17:44:50, Version 2, which is 11:44:50
local daylight savings time in Casper, Wyoming. The filename time refers to approximately the middle of
the measurement sequence. Totality in Casper started at 11:42:39 and ended at 11:45:05. Version 2
refers to the reprocessing of data with the latest CCD flat-fielding and stray-light corrections (Herman et
al., 2018; Marshak et al., 2018; Geogdzhayev and Marshak, 2017), and the geolocation algorithms.
The observing conditions for 21 August 2017 ranged from significant cloud cover over the oceans to
nearly clear skies over the United States (Figs. 1 and 2). The synoptic observations provided a unique
opportunity to estimate the fraction of reduced reflected radiation from the entire sunlit Earth caused
by a total solar eclipse.  Two of the synoptic observations were timed so that they centered on Casper,
Wyoming (42.8666° N, 106.3131° W, 17:44:50 UTC) and Columbia, Missouri (38.9517° N, 92.3341° W,
18:14:50 UTC).  Ten narrowband images were obtained at center vacuum wavelengths $\lambda_i$ of 317.5±0.5,
325±0.5, 340±1.3, 388±1.3, 443±1.3, 551±1.5, 680±0.8, 688±0.42, 764±0.5 and 779.5±0.9 nm (Herman
et al., 2018). Of these, 388, 443, 552, 680, and 779 nm radiances are not strongly absorbed in the
atmosphere and are used for estimating the reduction in reflected radiances from the Earth.  The others
are strongly affected either by ozone (317, 325, 340 nm) or oxygen absorption (688, 764 nm) in the
atmosphere, but give similar radiance percent reductions during the eclipse compared to non-absorbed
channels.
The non-absorbed wavelength observations were combined to produce eye-realistic color images
(https://epic.gsfc.nasa.gov). For this eclipse day study, 21 August, the original color images were
modified by increasing the gamma correction to better show the umbra over Casper, Wyoming and
Columbia, Missouri (Figs. 1 and 2 based on a suggestion by Steven Albers and Michael Boccara, 2017,
Private Communication). The images include Rayleigh scattering effects that cause light from the
penumbral region to increase illumination within the umbra along with scattering from clouds and
aerosols.

Table 1 summarizes eclipse timing and location details for Casper, Wyoming. During the 2.7
minutes needed to obtain all 10 wavelength channel images, the center of totality moves at about 46
km/minute or covering approximately 124 km. Based on the image in Fig. 1, the entire measurement
took place within the observed nearly clear-sky region surrounding Casper, Wyoming. A similar table
could be constructed for the eclipse totality region near Columbia, Missouri.

Table 1 *Eclipse Measurement Timing and Location Details for 5 Wavelengths*
Eclipse Maximum and EPIC Image Times.  Total Measurement Duration 2.7 minutes

| Wavelength (nm) | Date and Time | Location Name | Longitude |
|---|---|---|---|
|  | 2017-08-21 17:35:40 | Eclipse West Edge of WY state | $-111^{O}02'$ |
| 551 | 2017-08-21 17:42:36 | West of Casper | $-106^{O}22'$ |
| 680 | 2017-08-21 17:43:30 | West of Casper | $-106^{O}21'$ |
| Casper Wyoming | 2017-08-21 17:43:51 | Casper WY | $-106^{O}19'$ |
| 780 | 2017-08-21 17:44:24 | Near Glenrock WY | $-105^{O}52'$ |
| 443 | 2017-08-21 17:44:50 | West of Douglas WY | $-105^{O}14'$ |
| 388 | 2017-08-21 17:45:18 | West of Douglas WY | $-105^{O}17'$ |
|  | 2017-08-21 17:48:04 | Eclipse East Edge of WY state | $-104^{O}03'$ |


The timing and predicted shape of the Moon's shadow over Casper, Wyoming and Columbia,
Missouri can be seen at https://eclipse2017.nasa.gov/eclipse-maps. An annotated portion of the figures
for Casper and Columbia are reproduced in the Appendix (Fig. A1).  The predicted totality shadow (Fig.
A1) over Casper was elliptical in shape with a width of about 116 km (about $1.5^{O}$ in latitude or
longitude). The similar drawing for Columbia, Missouri shows a more nearly circular region of totality.
The dimension of the partial eclipse for 90 % obscuration is about $5^{O}$ in latitude or longitude.  The region
of 75 % obscuration covers a latitude range $32^{O}$ to $46^{O}$ or about 1200 km.  An obscuration region of this
size produces a significant decrease in the percentage of total solar irradiance reaching the Earth's
surface and in the amount reflected back to space. EPIC synoptically measures both the local and sunlit
portion of the global percent change in reflected radiance, which is approximately the same as the
percent change in global surface irradiance for the wavelength range from 388 to 780 nm. An exception
is within the umbral region, where the percent change is larger at the surface than at the top of the
atmosphere. The three wavelength channels shorter than 388 nm are affected by ozone absorption and
also do not contribute much to the sum of reflected radiances compared to the range from 388 to 780
nm. The energy content of 317 to 340 nm are not included in the quantitative estimate of broadband
(UV + visible) reduced reflected radiance, nor are the strongly absorbed $O_2$ A- and B-band channels, 688
and 764 nm, included. However, the effects of the eclipse on all 10 channels are individually estimated.

**2.1 Monochromatic Eclipse Images**

Before quantitatively examining the EPIC data from the eclipse in units of C/s or reflectance, the
same data can be represented as monochrome grey-scale images. The images (Fig. 3 with North down)
range from 340 nm, with strong Rayleigh scattering effects and some ozone absorption, to 780 nm in
the near infrared. North is selected as down to correspond to a 3D projection image presented later.
Because of the clarity of the atmosphere at 780 nm, the image serves as a geographic map of the Earth
as viewed by EPIC where North and South America are clearly visible.

**3.0 Results**
**3.1 Comparison of EPIC Observations of Eclipse Totality (21 Aug) with Non-Eclipse Days (20 and 23**
**Aug) for Casper, WY and Columbia, MO**
Atmospheric conditions during the eclipse at Casper, Wyoming were almost cloud-free
compared to Columbia, Missouri, which had optically thin low altitude clouds (Fig. 2). Figure 4 shows the
cloud cover on the day of the eclipse, 21 August 2017 (panel A) about 90 minutes before totality at
Casper and about 2 hours after totality. The eclipse umbra is still visible over the Atlantic Ocean. The
images (north is up) show that the skies remained relatively clear over the northern United States for
the duration of the eclipse. A similar set of images (panel B) are shown for the day before (20 August)
and two days after the eclipse (23 August). There was no useable data available on 22 August. Data
obtained on 20 and 23 Aug. at approximately the same UTC (backscatter phase angle for a given location
on Earth) as occurred during the total eclipse are used as reference data to compare with the eclipse
data on 21 Aug. The basic global patterns of cloud cover are similar for all three days, but not identical.
As shown later, the amount of light reflected back to space is approximately the same on the two non-
eclipse days 20 August and 23 August.
Figure 5 (upper panels A and B) shows longitudinal slices of 443 nm reflected solar radiances in
C/s towards $L_1$ through the locations corresponding to Casper, Wyoming and Columbia, Missouri at their
respective times of totality. The lower panels (C and D) of Fig. 5 show 443 nm measurements in C/s on
20 Aug at 18:04 UTC before the eclipse for nearly identical solar phase angles conditions for both sites.
The effect of clouds at the Columbia site compared to Casper can be seen in terms of the depth of the
umbra relative to the average C/s from $-140^O$ to $-150^O$ longitude (Panels A: ratio = 1530 and B: ratio =
37). Similarly, on the preceding day, 20 Aug (panels C and D), the cloud effect is small at Casper, $1.2 \times 10^4$
C/s, compared to Columbia, $5 \times 10^4$ C/s and just to the west of Columbia, $1.3 \times 10^5$ C/s.

The minimum 443 nm values during totality are 16.6 C/s for Casper and 312 C/s for Columbia.
On 20 Aug. EPIC measured 15240 C/s and 52728 C/s, respectively, showing the effect of increased
cloudiness for Columbia. While Fig. 5 is expressed in C/s, the data can be converted to radiance W/(m$^2$
nm sr) based on an in-flight determined radiance calibration coefficient of $K_R(443nm) = 5.291 \times 10^{-6}$
W/(m$^2$ nm sr C/s) derived from reflectance coefficients (Geogdzhayev and Marshak, 2017; Marshak et
al., 2018; Herman et al. 2018).  For 443 nm channel, an average count rate for the illuminated earth is
$3 \times 10^4$ C/s corresponding to a radiance of 0.159 W/(m$^2$ nm sr).  EPIC calibration constants for 8 of the 10
channels were obtained by in-flight comparisons of reflectance measured by two well calibrated low
Earth orbiting satellite instruments, OMPS (Ozone Mapping Profiler Suite for UV channels) and MODIS
(Moderate Resolution Imaging Spectroradiometer for visible and near-IR channels) for simultaneously
viewed Earth areas with the same satellite view and solar zenith angles (Herman et al., 2018;
Geogdzhayev and Marshak, 2017). The $O_2$ A- and B-band channels were calibrated using lunar data
when the Moon was within the field of view of EPIC. Detailed discussions and values of all EPIC
calibration coefficients K($\lambda$) are given by Geogdzhayev and Marshak (2017), Herman et al, (2018) and
Marshak et al., (2018). Most of the conclusions in this study are in terms of ratios of C/s from the same
wavelength channel at approximately the same solar phase angle that are independent of the absolute
calibration conversion from C/s to radiance.

The ratio $R_{EN}(\lambda_i)$ = I(20 August)/I(21 August) is used to characterize the eclipse effects at the top
of the atmosphere. Because the solar phase angles are nearly the same, the effects of the $172^O$
backscatter angle ("hot spot" caused mostly by minimized shadows) and ocean specular reflection are
also nearly the same on both days.
There is considerable variability in $R_{EN}(\lambda_i)$ as a function of wavelength that is partially caused by
the 2.7 minutes needed to obtain measurements for all 10 wavelengths. During the 2.7 minutes, the
center of totality moved about 124 km or about $1.7^O$ longitude, meaning that the ratio was affected by
atmospheric variability (mostly cloud effects) in the successive scenes containing the eclipse totality for
each wavelength.  The ratios $R_{EN}(\lambda_i)$ of C/s on the eclipse day to the preceding non-eclipse day are shown
in Fig. 6 for all 10 wavelength $\lambda_i$ channels and two sites (Casper, Fig 6a and Columbia, Fig. 6b) and
summarized in Table 2.  The same reference data from 20 Aug is used for both sites, since it was the
closest in UTC for both the Casper and Columbia eclipse times.


*Table 2 Maximum Radiance Ratio $R_{EN}(\lambda_i)$ during eclipse totality 17:44:50 UTC (Casper) and   18:14:50 UTC (Columbia) compared to 20 Aug. at 18:03:59 for both sites (see Fig. 6).*

| Wavelength $\lambda_i$ (nm) | Max. $R_{EN}(\lambda_i)$ C/s Casper, Wyoming | Max. $R_{EN}(\lambda_i)$ C/s Columbia, Missouri |
|---|---|---|
| 317.5 | 255 | 50 |
| 325 | 245 | 49 |
| 340 | 475 | 59 |
| 388 | 3500 | 81 |
| 443 | 935 | 169 |
| 551 | 5455 | 183 |
| 680 | 220 | 171 |
| 688 | 365 | 246 |
| 764 | 302 | 92 |
| 780 | 395 | 38 |

For the eclipse study, the range of synoptically observed longitudes is approximately from the
international dateline (-180$^O$) to almost longitude of Greenwich, England (0$^O$).  The nearly clear-sky in
Casper with optically thin clouds permits the reflected light during totality to become very small (about
17 C/s for 443 nm compared to 1.5x10$^4$ C/s on 20 August at the same longitude). Columbia had more
low altitude cloud cover than Casper (Fig. 2) with the cloud cover extending into the region of totality.
The effect of this cloud cover can be seen in Fig. 6, where the maximum $R_{EN}$(443, Columbia) = 169
compared to 935 for Casper. Table 2 provides the eclipse radiance ratio $R_{EN}(\lambda_i)$ for the five non-absorbed
wavelength and 5-absorbed channels that can help validate 3D radiative transfer models. The measured
lower values $R_{EN}(\lambda_i)$ at Columbia compared to Casper show that there is high sensitivity in the TOA
upwelling measured ratios to the presence of even optically thin clouds. A detailed radiative transfer
study for realistic conditions is made feasible by using EPIC's simultaneous estimates of cloud reflectivity
and transmission, cloud height, ozone amounts, (Fig. A3 and Herman et al., 2018), and aerosol amounts
(Torres et al., 2018 private communication).   These data products are available from the NASA-Langley
data repository referenced above.
## 3.2 Global reduction of reflected sunlight during the eclipse over Casper WY
The unique DSCOVR/EPIC measurements provide estimates of the fractional reduction of
sunlight from 388 to 780 nm reflected back to space for the entire sunlit globe caused by the eclipse
shadow on the Earth. To do this, all of the light reaching EPIC in each of the five non-absorbed channels,
388, 443, 551, 680, and 780 nm, are integrated over the visible sunlit Earth and compared (percent
difference PDF($\lambda_i$)) with a nearly identical viewing geometry (nearly the same UTC) from the previous
and next days. The assumption is that the major cloud features change slowly on a global scale over
relatively short periods (Figs. 1 to 3).  A test of this hypothesis is that the PDF between successive non-
eclipse days is small compared to the eclipse day PDF($\lambda_i$) with a non-eclipse day.
In the 3D Fig. 7 for 443 nm, the nearly cloud free eclipse region is the blue area in the midst of
greens, yellows, and reds.  The high red values correspond to fairly reflective clouds mostly seen near
the equator (Fig. 1). The yellows and greens correspond to lower altitude clouds that tend to have
smaller reflectivities.  Integrating over all of the pixels for the eclipse on 21 August 2017, using the file
named epic_1b_20170821174450_02.h5, we get S(DOY, UTC) = 5.34366x10$^{10}$ C/s for DOY=233 (21
August 2017) and UTC=17:44:50.  For the eclipse day, the 443 nm average C/s = 2.0631x10$^4$, which
corresponds to 2.0631x10$^4$ $K_R$(443 nm) = 0.11 W/(m$^2$ nm sr).  Peak values are approximately 1x10$^5$ C/s,
or about 0.53 W/(m$^2$ nm sr).  Figure 7 is oriented with north down so as to be able to see into the eclipse
shadow region. A similar figure is obtained for Columbia, Missouri with reduced depth caused by some
visible light cloud cover extending into the region of totality (Fig. 2).

Measured C/s images for six wavelength channels (340 to 780 nm) on 20, 21, and 23 August (Fig.
8) were selected to be as close as possible to the UTC time of the eclipse in Casper Wyoming, keeping
the scattering phase angles nearly constant. Similar images for the strongly absorbed channels 317.5,
325, 688, 764 nm channels are shown in the appendix (Fig. A2). The middle images in panels B and E of
Figs. 8a, 8b and 8c are for the eclipse over Casper, Wyoming. These images are in the same format as
Fig. 3, but rotated with north up. Unlike Fig. 3, the scale in Fig. 8 was selected so that the brightest
clouds do not saturate the image. The increase in scale makes the land surfaces less visible. While the
figures are similar from wavelength to wavelength, there are differences in the depth of the eclipse
totality and the reflectivities of the surrounding clouds. In general, the equatorial clouds with higher C/s
(reflectivities) tend to reach higher altitudes. This is confirmed by examining the C/s in the strongly
absorbed $O_2$ A-band channel (Fig. A2b and Herman et al., 2018).
EPIC measured $C(\lambda)$ in C/s for each pixel can be converted to Earth top of the atmosphere
reflectance $Re(\lambda)$ using the in-flight derived calibration coefficients $K(\lambda)$, where  $Re(\lambda) = K(\lambda)\, C(\lambda)$. For
the six wavelength channels in Fig. 8 plus the $O_2$ A- and B-band channels, $K(340) = 1.975 \times 10^{-05}$, $K(388) =$
$2.685 \times 10^{-05}$, $K(443) = 8.340 \times 10^{-06}$, $K(551) = 6.66 \times 10^{-06}$, $K(680) = 9.30 \times 10^{-06}$, $K(687.75) = 2.02 \times 10^{-05}$, $K(764)$
$= 2.36 \times 10^{-05}$, and $K(780) = 1.435 \times 10^{-05}$ (Herman et al., 2018; Geogdzhayev and Marshak, 2018; Marshak
et al., 2018).  To estimate the percent reduction in outgoing radiances, the ratios of integrals over the
illuminated CCD for each wavelength channel are formed for nearly the same Earth geometry on days
preceding and following the eclipse. Either the integrated reflectances or the integrated C/s x $10^{-7}$ (Eqn.
1) for Tables 3A for Casper, Wyoming and 3B for Columbia Missouri) over the CCD pixels, $ICs(\lambda)$, can be
used directly, since they are linearly proportional to the integral of the photons received by the
illuminated pixels.
Table 3 and Fig. 9 show that the global reduction of backscattered light caused by the eclipse is
similar for the two sites even though there is more cloud cover locally over Columbia than Casper. This is
because the global reduction caused by the differing umbral regions is a small fraction of the total, and
only 30 minutes have elapsed between the two measurements, which is not enough time for the global
cloud cover to have significantly changed.

Table 3A Global integral of reflected light ICs for the UTC of the Casper, WY eclipse on 21 August and for the closest solar phase angle from 20 and 23 August. PDF is the percent difference caused by the eclipse. Units are ICs x $10^{-7}$

| $\lambda_i$ (nm) | 20 August 2017 16:58:31 UTC | 21 August 2017 17:44:50 | 23 August 2017 17:54:36 | Avg. PDF |
|---|---|---|---|---|
| 317.5 | 280.5 | 258.8 | 282.0 | 9±0.3 |
| 325 | 460.6 | 425.5 | 464.2 | 9±0.4 |
| 340 | 3183 | 2946 | 3213 | 9±0.5 |
| 388 | 2034 | 1878 | 2044 | 9±0.3 |
| 443 | 5808 | 5344 | 5813.2 | 9±0.05 |
| 551 | 5619 | 5078 | 5573 | 10±0.5 |
| 680 | 3790 | 3433 | 3773 | 10±0.3 |
| 688 | 1129 | 1010 | 1110 | 11±0.9 |
| 764 | 671.9 | 585.9 | 651.9 | 13±1.7 |
| 780 | 2794 | 2491 | 2799 | 12±0.1 |


Table 3B Global integral of reflected light ICs for the UTC of the Columbia, MO eclipse on 21 August and for the closest solar phase angle from 20 and 23 August. PDF is the percent difference caused by the eclipse. Units are ICs x $10^{-7}$

| $\lambda_i$ (nm) | 20 August 2017 18:03:359 GMT | 21 August 2017 18:14:50 | 23 August 2017 17:54:36 | Avg. PDF |
|---|---|---|---|---|
| 317.5 | 281.3 | 258.3 | 282.0 | 9±0.1 |
| 325 | 461.6 | 425.9 | 464.2 | 9±0.3 |
| 340 | 3193 | 2956 | 3213 | 8±0.3 |
| 388 | 2034 | 1884 | 2044 | 8±0.3 |
| 443 | 5813.7 | 5372.3 | 5813.2 | 8±0.01 |
| 551 | 5586 | 5091 | 5573 | 10±0.1 |
| 680 | 3790 | 3453 | 3773 | 10±0.2 |
| 688 | 1121 | 1011 | 1110 | 10±0.5 |
| 764 | 661.2 | 576.0 | 651.9 | 14±0.8 |
| 780 | 2794 | 2475 | 2799 | 13±0.1 |

Figure 9 shows a plot of the data contained in Table 3 based on Eqn. 1. The two non-eclipse days are
nearly identical, while the eclipse day (21 Aug) is significantly lower at all wavelengths. The
backscattered light (in C/s) peaks near 500 nm and then decreases toward longer wavelengths, since
C($\lambda$) is proportional to the solar irradiance, which decreases with $\lambda$ after approximately 550 nm.

$$ICs(\lambda) = \int_0^{2048} \int_0^{2048} C(\lambda, x, y)dxdy \qquad (1)$$

over Np = 2.59x$10^6$ illuminated pixels on 20, 21, 23 August 2017



For the 443 nm channel, the result is an approximate decrease of 9 % on 21 August at 11:44:50
for Casper and 8% at 12:14:50 for Columbia local time.  As a reference, we compare two non-eclipse
days (20 and 23 August). The relative difference is (5808-5813)/5813 0.1 % for Casper and 0.01 % for
Columbia, which is much smaller than the 9 % decrease produced by the eclipse on 21 August. The
comparison of the 443 nm eclipse day with two non-eclipse days gives a measure of the uncertainty in
the calculation (e.g., 9 ± 0.05 % for Casper and 8 ± 0.01% for Columbia ).

Percent difference PDF($\lambda_i$) calculations for $\lambda_i$ = 317.5, 325, 340, 388, 443, 551, 680, 688, 764, and
780 nm, based on Eqn. 1 are summarized in Table 3A, yielding PDF($\lambda_i$) = 9, 9, 9, 9, 9, 9, 10, 10, 13, and 12
% reductions in backscattered radiances in the direction of $L_1$, respectively for Casper with similar values
for Columbia. The PDF(764 nm) within the strongly absorbing $O_2$ A-band is 13 % for Casper and 14% for
Columbia, even though the reflected ICs(764nm) is much lower than the surrounding non-absorbed
bands. The fact that adjacent absorbed and non-absorbed wavelengths give consistent PDF($\lambda_i$) suggests
that most of the effect comes from clouds. Eclipse effects for the short UV wavelengths are affected by
Rayleigh scattering and clouds, and not much by the relatively low UV surface reflectivity (about 4%).
Eclipse effects on outgoing radiances for wavelengths longer than about 700 nm are increased by
vegetation reflectivity, even where the amount of clear-sky penetrating radiances are small for the O2

688 and 764 nm channels. There is insufficient information to explain the small observed wavelength
dependence in Table 3.

To estimate the fractional reflected radiance reduction for the wavelength range from 388 to
780 nm, a polynomial interpolation $R(\lambda)$ of the Avg. PDF in Table 3 for the 5 weakly absorbed channels is
formed (Fig. 10 panels A and B red curves). $R(\lambda)$ must be weighted by the solar irradiance spectrum $F(\lambda)$.
The solar spectrum used is a combination of measured solar flux data named "atlas_plus_modtran" in
the libRadtran software package (Mayer and Kylling, 2005). The components, $F_R$ and $F_S$, of the weighted
average R are defined in Eqns. 2 and 3. On 21 August 2017 the distance of the Earth from the Sun was
1.011 AU, or $F_S$(21 Aug at 1 AU) = 664.94 $W/m^2$ and at 1 AU, $F_{R-Casper}$ = 66.11 $W/m^2$ and $F_{R-Columbia}$ = 64.86
$W/m^2$. For the wavelength range of interest (387.9 to 781.25 nm), $F_S$ is about half of the total solar
irradiance of 1361 $W/m^2$ at the top of the atmosphere at 1 AU (Kopp and Lean, 2011).


$$F_S = \int_{387}^{781} F(\lambda)d\lambda \qquad\qquad F_R = \int_{387}^{781} R(\lambda)F(\lambda)d\lambda \qquad (2)$$

$$<R> = \frac{\int_{387}^{781} R(\lambda)F(\lambda)d\lambda}{\int_{387}^{781} F(\lambda)d\lambda} \qquad\qquad \begin{aligned} <R_{Casper}> &= 0.101 \\[1em] <R_{Columbia}> &= 0.098 \end{aligned} \qquad (3)$$

Figure 11b shows the product $R(\lambda)F(\lambda)/F_S$ ($nm^{-1}$). Forming <R> shows that during the eclipse the
shadow of the Moon reduces the backscattered radiance (388 to 780 nm) from the sunlit Earth in the
direction of $L_1$ by about 10 %. The combined uncertainty ±0.3 % is caused by variations in the cloud
cover of the reference days compared the eclipse day. The calculation of <R> is based on C/s
measurements from DSCOVR/EPIC of the sunlit Earth and the interpolation function $R(\lambda)$. The result is
independent of the absolute calibration of EPIC, since it is based on ratios of C/s over three days with
approximately the same UTC (scattering phase angles). $R(\lambda)$ includes the near backscatter direction
enhanced reflection function appropriate for the entire sunlit disk at a backscatter angle of about $172^O$.
The three days at nearly the same UTC can be compared directly, since EPIC has proven to be very stable
based on repeated in-flight calibrations over a 2-year period using OMPS and MODIS (Herman et al.,
2018 and Geogdzhayev and Marshak, 2018). The smooth function $R(\lambda)$ does not include absorption
features from water and the $O_2$ A- and B-bands.

 **3.3 Comparison of EPIC albedo with POLDER reflectance**

The TOA albedo measurements made by EPIC can be compared with reflectance measurements
made by the POLDER satellite instrument near the hotspot backscatter direction (172$^O$) for the incident
solar irradiance over nearly cloud-free scenes (Maignan et al., 2004). EPIC C/s can be converted to
albedo using the calibration constants K($\lambda$), which already contains the factor $\pi$ (Fig. 11A). The average
TOA albedo from EPIC was almost the same on 20 Aug. as on 23 Aug. For EPIC albedo data over
grassland common to Casper, Wyoming compared to the POLDER measurements, the C/s data for each
wavelength (see Fig.5 for 443 nm) can be converted to TOA albedo.

Measurements from the POLDER satellite over Khingan Range, China (117.55°E to131.56°E,
45.68°N to 53.56°N) show that the backscatter amount from the land surface increases with increasing
wavelength (Maignan et al., 2004). The Khingan range is mainly covered by deciduous broadleaf and a
mix of deciduous and evergreen needle leaf forest with a small amount of grassland, while the area
around Casper is mainly short grass prairie land with few trees. Over Casper, WY (Fig. 11B), the
wavelength dependence of the EPIC TOA albedo (551, 680, and 780 nm) at 172$^O$ backscatter angle is
similar to POLDER surface reflectance at 8$^O$.  The shape and magnitude differences are partially caused
by the atmospheric component of the albedo that includes some light cloud cover, whereas the POLDER
reflectance has atmospheric effects subtracted.   The effect of increasing Rayleigh scattering is seen for
shorter wavelengths measured by EPIC.

**3.0 Summary**

The EPIC instrument onboard the DSCOVR spacecraft synoptically observes the entire sunlit
portion of the Earth from an orbit near the Earth-Sun Lagrange-1 point. On 21 August 2017, EPIC was
able to observe the totality shadow from the lunar eclipse of the Sun with the Earth's surface for about 3
hours (seven 10-channel measurements) as it crossed the United States from west to east (about 1.5
hours).  When the region of totality was over Casper, Wyoming at 17:44:50 UTC, the reflected 443 nm
TOA radiance was reduced to 16 C/s (8x10$^{-5}$ W/m$^2$sr) in the narrow region of totality compared to a non-
eclipse day (1.52x10$^4$ C/s or 0.076 W/m$^2$sr). About 30 minutes later the shadow passed over Columbia,
Missouri, but the presence of thin clouds in the vicinity of Columbia caused increased reflected radiance
of 312 C/s (1.6x10$^{-3}$ W/m$^2$sr) into the umbral region during totality compared to Casper.  The ratio $R_{EN}(\lambda_i)$
of reflected radiances within the eclipse totality to radiances for the same geometry on adjacent non-
eclipse days was measured for all 10 wavelength channels. The measured $R_{EN}$(443 nm) was smaller for
Columbia (71) than for Casper (936), showing the sensitivity to increased cloud cover over Columbia.
Similarly $R_{EN}$(388 nm, Casper) = 3500 and  $R_{EN}$(388 nm, Columbia) = 81.    While the results cannot be
directly compared with $R_{EN}$, good agreement was obtained (Kazantzidis et al., 2007) between a model
study based on a 3D Monte Carlo radiative transfer model (Emde and Mayer, 2007) and measured ratio
at 380 nm (ratio = 217) of downward global surface radiation before and during totality. The measured
radiance ratios $R_{EN}(\lambda_i)$ can serve as a validation data set for 3D radiative transfer models of the
atmosphere that include cloud effects, since EPIC also measures the surrounding amount of cloud cover
for the entire sunlit Earth. Comparing $R_{EN}(\lambda$, Casper) with $R_{EN}(\lambda$, Columbia) shows that Rayleigh
scattering combined with low optical depth clouds can scatter light into the umbra region and reduce
$R_{EN}(\lambda)$. Outside of the region of totality, EPIC observed the partial eclipse shadow and the fully
illuminated regions of the Earth's disk. Interpolating in wavelength between the percent reductions in
integrated radiances (in C/s) over the sunlit globe, $ICs(\lambda_i)$ for the 5 measured non-absorbed wavelength
channels at both locations showed that the integrated reflected radiance from the Earth's sunlit disk
towards $L_1$ decreased by about 10 % compared to the integrated radiances measured on the days before
and after the eclipse for approximately the same observing geometry as occurred during the eclipse.
Similar calculations comparing two non-eclipse days show smaller changes in $ICs$ (less than 0.1 %) than
the eclipse-day change. The five channels that are partially absorbed in the atmosphere give consistent
results compared to the non-absorbed channels suggesting that cloud reflectivities dominate the 317.5
to 780 nm radiances reflected back to space from the sunlit Earth's disk with a contribution from
Rayleigh scattering for the shorter wavelengths.

## Appendix

The course of the eclipse in the vicinity of Casper, Wyoming and Columbia, Missouri is shown in Fig. A1

Greyscale images for the short UV wavelength channels (317.5, 325) with strong ozone absorption and Rayleigh scattering, the longer wavelength UV channels (340, 388), and the strongly absorbed $O_2$ B- and A-band channels (688, 764 nm) are shown in Figs. A2a, A2b, A2c

The amount of ozone over the eclipse sites can be derived (Herman et al., 2018) to produce ozone data that is stored in the NASA-Langley archive. During the eclipse, it is not possible to derive the amount of ozone from either ground-based or satellite data. Ozone amounts do not change rapidly from day to day except when major weather systems pass through a region, which was not the case during the eclipse period, 20 August to 23 August. This is confirmed from OMI satellite data (Ozone Monitoring Instrument onboard the AURA satellite). Figure A3 shows the amount of ozone over the eclipse trajectory obtained on 20 August. The values obtained 316 DU near Casper, WY and 306 DU near Columbia compare well with ozone amounts derived from OMI of 314 DU and 301 DU. The $O_3$ variability during the 2.7 minutes (approximately 124 km or about $1^O$ of longitude) is about ±5 DU.

### 4.0 Author Contributions

Jay Herman wrote most of the paper and performed most of the calculations

Guoyong Wen is the funded principal investigator of the project.

Alexander Marshak provided the calibration coefficients for the visible and near-IR channels

Karin Blank provided the color images in Figs. 1 to 3. She was responsible for the geolocation of the 10 filter images on a common grid.

Liang Huang provided the calibration coefficients for the UV channels

Alexander Cede provided the flatfielding, stray light correction, and dark current analysis

Nader Abuhassan helped with flatfielding and stray light correction and was responsible for the ground-based portion of this research.

Matthew Kowalewski provided the flatfielding, stray light correction, and dark current analysis

The authors declare that they have no conflict of interest.

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

Acknowledgement
The author would like to thank the DSCOVR project for support in completing this study as well as
financial support from an accepted NASA-ROSES proposal in response to NNH16ZDA001N-ISE. All data is
from the permanent NASA data repository:
https://eosweb.larc.nasa.gov/project/dscovr/dscovr_epic_l1b.

**Tables**

Table 1  *Eclipse Measurement Timing and Location Details for 5 Wavelengths*

Eclipse Maximum and EPIC Image Times.  Total Measurement Duration 2.7 minutes

| Wavelength (nm) | Date and Time | Location Name | Longitude |
|---|---|---|---|
|  | 2017-08-21 17:35:40 | Eclipse West Edge of WY state | -111$^{O}$02' |
| 551 | 2017-08-21 17:42:36 | West of Casper | -106$^{O}$22' |
| 680 | 2017-08-21 17:43:30 | West of Casper | -106$^{O}$21' |
| Casper Wyoming | 2017-08-21 17:43:51 | Casper WY | -106$^{O}$19' |
| 780 | 2017-08-21 17:44:24 | Near Glenrock WY | -105$^{O}$52' |
| 443 | 2017-08-21 17:44:50 | West of Douglas WY | -105$^{O}$14' |
| 388 | 2017-08-21 17:45:18 | West of Douglas WY | -105$^{O}$17' |
|  | 2017-08-21 17:48:04 | Eclipse East Edge of WY state | -104$^{O}$03' |




Table 2 *Radiance Ratio $R_{EN}(\lambda_i)$ during eclipse*
*totality 17:45 UTC compared to 20 Aug*

| Wavelength $\lambda_i$ (nm) | Max. $R_{EN}(\lambda_i)$ C/s |
|---|---|
| 317.5 | 118 |
| 325 | 68.2 |
| 340 | 144 |
| 388 | 86 |
| 443 | 122 |
| 551 | 119.5 |
| 680 | 80 |
| 688 | 38 |
| 764 | 108 |
| 780 | 112.5 |









*Table 3A Eclipse change in reflected light at Casper, WY from 20, 21, 23 August 2017 Units are ICs x 10[-7]*

| $\lambda_i$ (nm) | 20 August 2017 16:58:31 GMT | 21 August 2017 17:44:50 | 23 August 2017 17:54:36 | Avg. PDF |
|---|---|---|---|---|
| 317.5 | 280.5 | 258.8 | 282.0 | 9±0.3 |
| 325 | 460.6 | 425.5 | 464.2 | 9±0.4 |
| 340 | 3183 | 2946 | 3213 | 9±0.5 |
| 388 | 2034 | 1878 | 2044 | 9±0.3 |
| 443 | 5808 | 5344 | 5813.2 | 9±0.05 |
| 551 | 5619 | 5078 | 5573 | 10±0.5 |
| 680 | 3790 | 3433 | 3773 | 10±0.3 |
| 688 | 1129 | 1010 | 1110 | 11±0.9 |
| 764 | 671.9 | 585.9 | 651.9 | 13±1.7 |
| 780 | 2794 | 2491 | 2799 | 12±0.1 |


*Table 3B Eclipse change in reflected light at Columbia, MO from 20, 21, 23 August 2017 Units are ICs x 10[-7]*

| $\lambda_i$ (nm) | 20 August 2017 18:03:359 GMT | 21 August 2017 18:14:50 | 23 August 2017 17:54:36 | Avg. PDF |
|---|---|---|---|---|
| 317.5 | 281.3 | 258.3 | 282.0 | 9±0.1 |
| 325 | 461.6 | 425.9 | 464.2 | 9±0.3 |
| 340 | 3193 | 2956 | 3213 | 8±0.3 |
| 388 | 2034 | 1884 | 2044 | 8±0.3 |
| 443 | 5813.7 | 5372.3 | 5813.2 | 8±0.01 |
| 551 | 5586 | 5091 | 5573 | 10±0.1 |
| 680 | 3790 | 3453 | 3773 | 10±0.2 |
| 688 | 1121 | 1011 | 1110 | 10±0.5 |
| 764 | 661.2 | 576.0 | 651.9 | 14±0.8 |
| 780 | 2794 | 2475 | 2799 | 13±0.1 |





**Figure Captions**
Fig. 1 Synoptic view of the sunlit Earth perturbed by the 21 August 2017 total eclipse centered over
Casper, Wyoming at 17:44:50 UTC. The black region is the eclipse umbra centered over Casper, WY. The
color image has been adjusted from the images on https://epic.gsfc.nasa.gov/ by increasing the gamma
correction (Cescatti, 2007) to bring out the region of totality and surrounding clouds.
Fig. 2 Synoptic view of the total eclipse centered over Columbia, Missouri at 18:14:50 UTC. The black
region is the eclipse umbra centered over Columbia, MO. The color image has been adjusted from the
images on https://epic.gsfc.nasa.gov/ by increasing the gamma correction to bring out the region of
totality and surrounding clouds.
Fig. 3 Greyscale images for 6 of the DSCOVR/EPIC channels for the eclipse over Casper Wyoming
showing the blurring caused by Rayleigh scattering and the dark land and ocean surfaces at 340 nm to
the almost clear atmosphere and bright continental surfaces at 780 nm. The images were obtained over
a period of 2.7 minutes. North is facing down. The greyscale is linear, with black representing very low
reflectivity and white very high reflectivity from high altitude equatorial region clouds.
Fig. 4 Panel A: Synoptic natural color images on 21 August at 16:14 and 19:44 before and after the
eclipse over the US, and Panel B: the days before and after the eclipse selected to be as close as possible
to the phase angle (UTC 17:44:50) as the time of totality over Casper, Wyoming. North is facing up.
Fig. 5 Top: The effect of an eclipse (21 Aug) on the measured C/s reflected back to space as a function of
longitude (degrees) for two locations, Casper Wyoming (left) and Columbia Missouri (right).  Bottom:
Measured C/s reflected back to space on 20 Aug. A $\log_{10}$ scale is used to show details of the spatial
variability mostly caused by clouds
Fig. 6a   The ratio $R_{EN}(\lambda_i)$ = I(Aug20)/I(Aug21) at the time of the Eclipse in Casper Wyoming for
wavelengths 317.5 to 780 nm. The channels 317.5 to 340 nm are affected by ozone absorption and the
channels 688  and 764 nm are within the $O_2$ B and A absorption bands.
Fig. 6b The ratio $R_{EN}(\lambda_i)$ = I(Aug20)/I(Aug21) at the time of the Eclipse in Columbia, Missouri for
wavelengths 317.5 to 780 nm. The channels 317.5 to 340 nm are affected by ozone absorption and the
channels 688  and 764 nm are within the $O_2$ B and A absorption bands.
Fig. 7 The C/s observed by EPIC for the 443 nm channel corresponding to the color image shown in Fig.
1. In the data file, the word infinity has been replaced by the number zero. In this image there are
approximately Np = $2.59 \times 10^6$ illuminated pixels out of $2048^2$ = $4.194304 \times 10^6$ pixels (61.8 %).
Fig. 8a Image in C/s for 340 and 388 nm for 20 Aug.(A+D), 21 Aug. (B+E), and 23 Aug. (C+F). The scale
applies to the specific wavelength. North is up.
Fig. 8b Image in C/s for 443 and 551 nm for 20 Aug.(A+D), 21 Aug. (B+E), and 23 Aug. (C+F). The scale
applies to the specific wavelength. North is up.
Fig. 8c Image in C/s for 680 and 780 nm for 20 Aug.(A+D), 21 Aug. (B+E), and 23 Aug. (C+F). The scale
applies to the specific wavelength. North is up.
Fig. 9  Average reflected light in C/s for eclipse (21 Aug. red) and non-eclipse (20 Aug. and 23 Aug. (black
and grey) days from Table 3 and Eqn. 1 for Casper and Columbia. The locations of the maxima are from
curve fitting to the discrete wavelength measurements.
Fig. 10 Solar Irradiance at 1 AU F($\lambda$) Watts/(m$^2$ nm)  (Mayer and Kylling, 2005) and the eclipse reduction
function R($\lambda$) in percent for Casper, Wyoming (red curve in panel A) and Columbia, Missouri (red curve
in panel B).  Fractional reduction (nm$^{-1}$) in reflected solar irradiance in the direction of L-1 for Casper,
Wyoming (panel C) and Columbia, Missouri (panel D)
Fig. 11 A. The measured albedo at Casper Wyoming on 20 Aug (black curve) and 23 Aug (grey curve)
compared to B the POLDER measured surface reflectance in the Khingan Range, China ( Maignan et al.,
2004) corresponding to 8$^O$ from overhead sun.
Fig. A1 The timing and shape of the Moon's shadow over Casper, Wyoming showing the relative location
of Casper and Columbia (white circles) at 11:45 MDT (Mountain Daylight Time) and 1:15 CDT (Central
Daylight Time). The shadow is moving at about 46 km/minute. (https://eclipse2017.nasa.gov/eclipse-
maps). The scale size with the NASA logo is 50 km.
Fig. A2a Image in C/s for 317 and 340 nm for 20 Aug., 21 Aug. and 23 Aug. The scale applies to the
specific wavelength. North is up.
Fig. A2b Image in C/s for 340 and 388 nm for 20 Aug.(A+C), 21 Aug. (B+E), and 23 Aug. (C+F). The scale
applies to the specific wavelength. North is up.
Fig. A2c A2c Image in C/s for 688 and 764 nm for 20 Aug., 21 Aug. and 23 Aug. The scale applies to the
specific wavelength. North is up.
Fig. A3 EPIC measured ozone amounts from 20 August in the vicinity of Casper, WY and Columbia, MO.

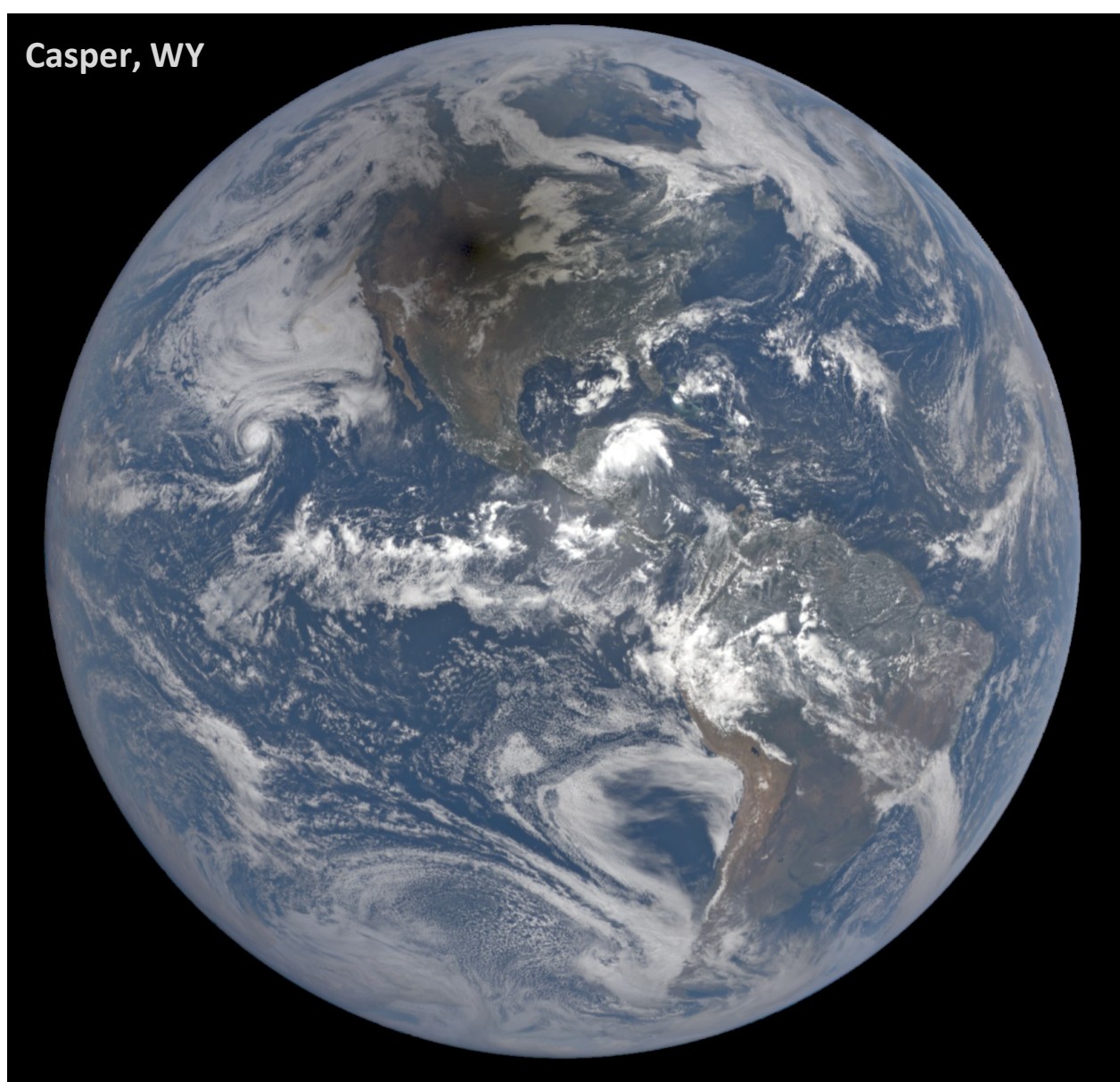

Fig. 1 Synoptic view of the sunlit Earth perturbed by the 21 August 2017 total eclipse centered over Casper, Wyoming at 17:44:50 UTC. The black region is the eclipse umbra centered over Casper, WY. The color image has been adjusted from the images on https://epic.gsfc.nasa.gov/ by increasing the gamma correction (Cescatti, 2007) to bring out the region of totality and surrounding clouds.



**F01**


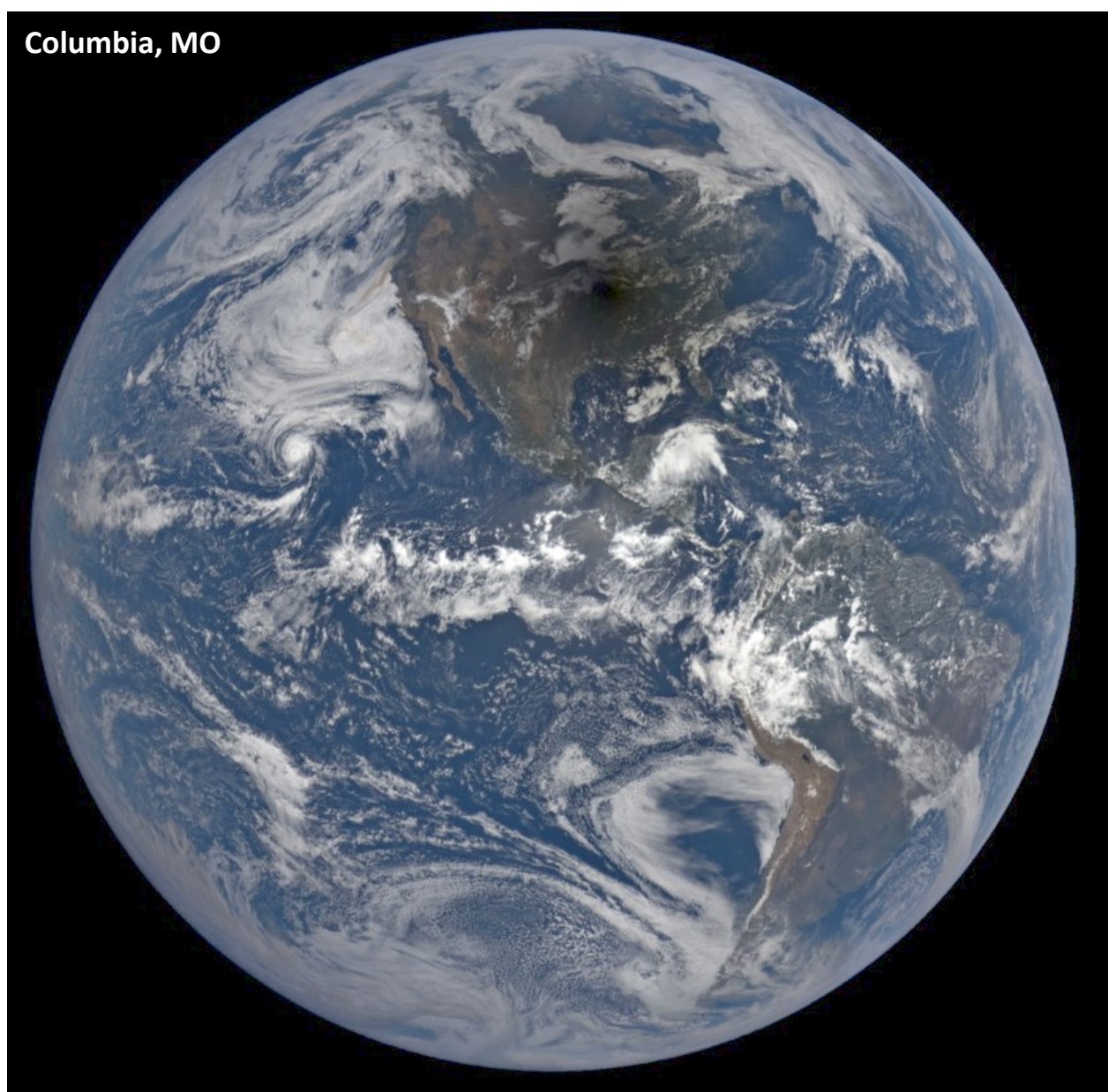

Fig. 2 Synoptic view of the total eclipse centered over Columbia, Missouri at 18:14:50 UTC. The black region is the eclipse umbra centered over Columbia, MO. The color image has been adjusted from the images on https://epic.gsfc.nasa.gov/ by increasing the gamma correction to bring out the region of totality and surrounding clouds.



**F02**


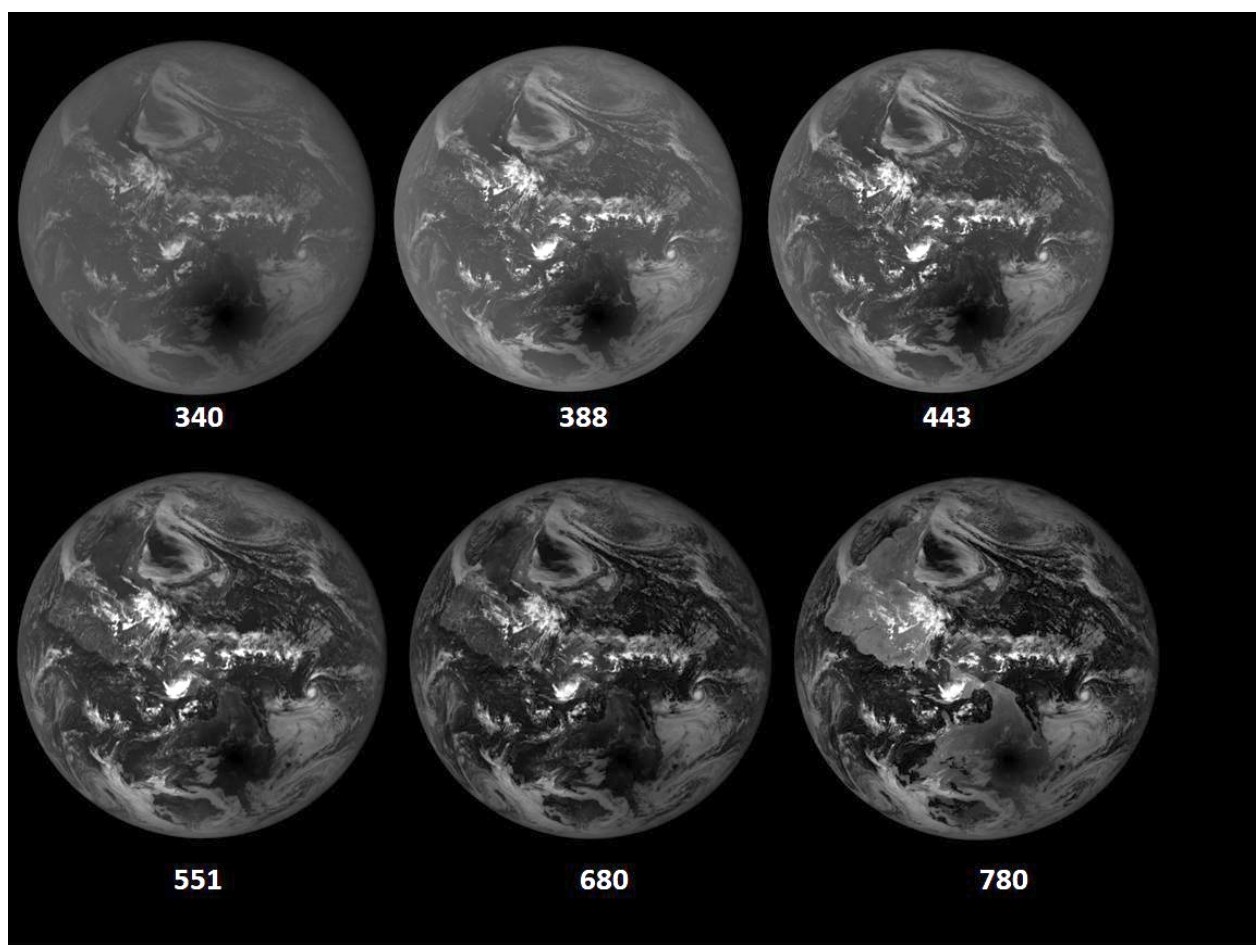

Greyscale images for 6 of the DSCOVR/EPIC channels for the eclipse over Casper Wyoming showing the blurring caused by Rayleigh scattering and the dark land and ocean surfaces at 340 nm to the almost clear atmosphere and bright continental surfaces at 780 nm. The images were obtained over a period of 2.7 minutes. North is facing down. The greyscale is linear, with black representing very low reflectivity and white very high reflectivity from high altitude equatorial region clouds.

**F03**

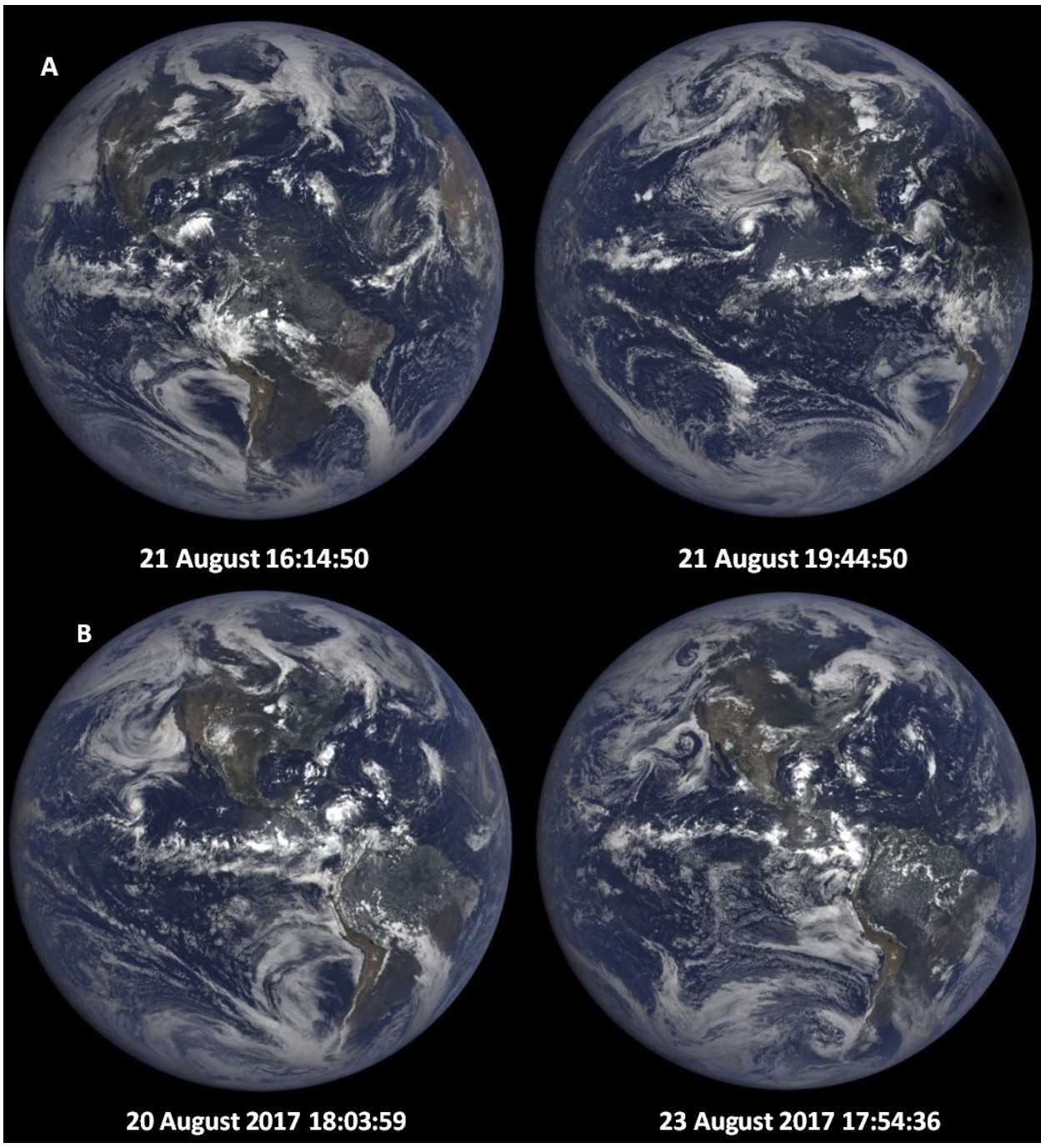

21 August 16:14:50        21 August 19:44:50

20 August 2017 18:03:59        23 August 2017 17:54:36

Fig. 4 Panel A: Synoptic natural color images on 21 August at 16:14 and 19:44 before and after the eclipse over the US, and Panel B: the days before and after the eclipse selected to be as close as possible to the phase angle (UTC 17:44:50) as the time of totality over Casper, Wyoming. North is facing up.


**F04**

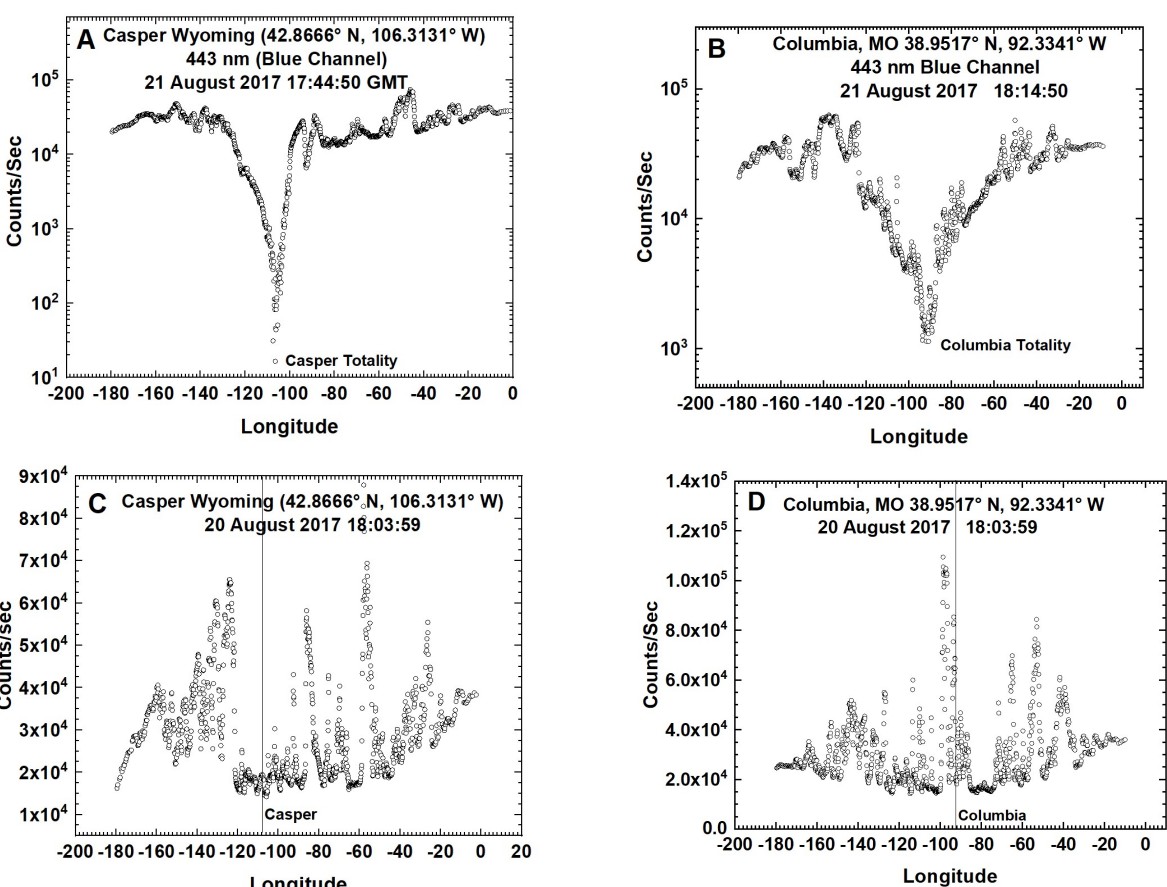

Fig. 5 Top: The effect of an eclipse (21 Aug) on the measured C/s reflected back to space as a function of longitude (degrees) for two locations, Casper Wyoming (left) and Columbia Missouri (right). Middle: Measured C/s reflected back to space on 20 Aug.



**F05**

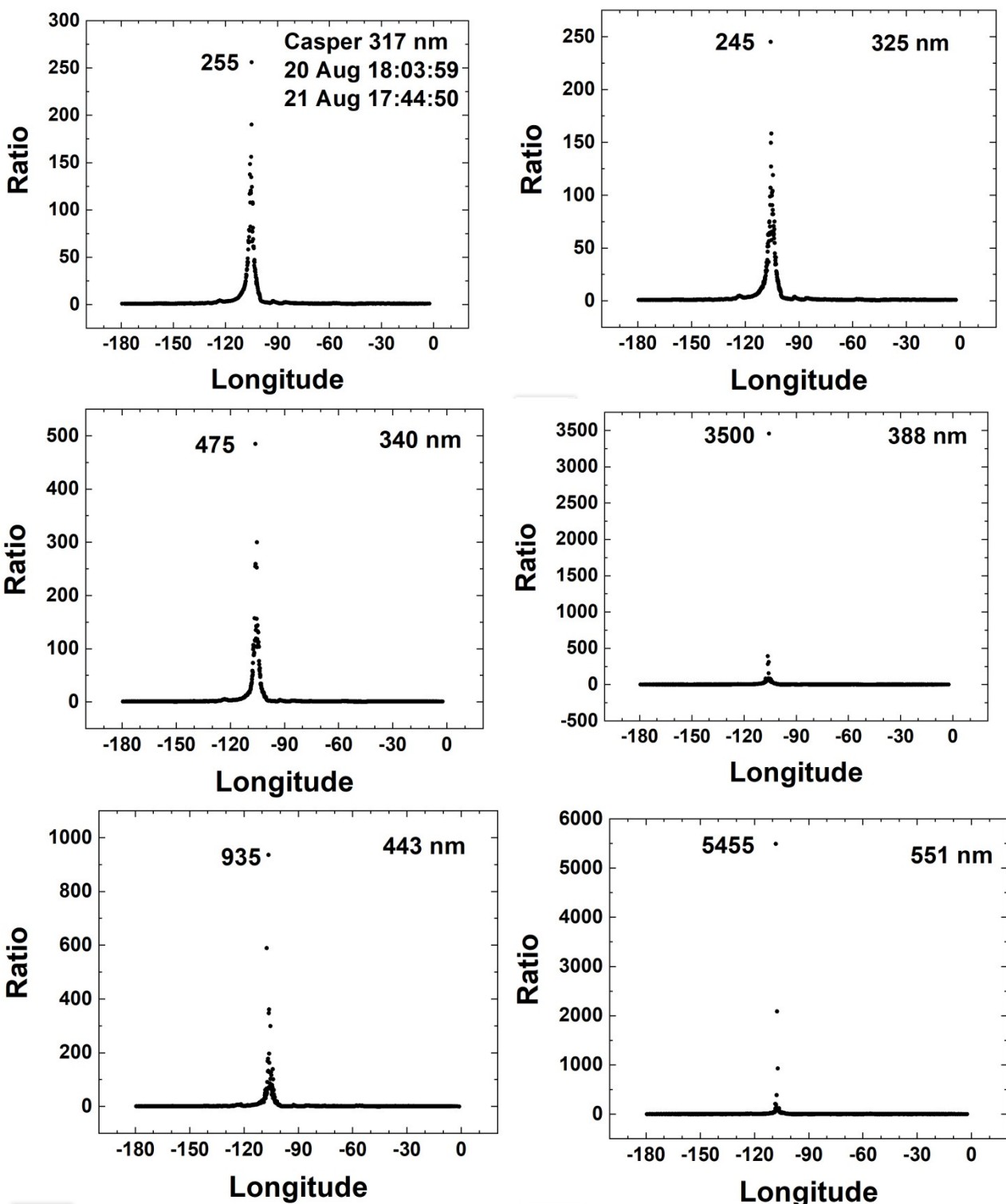


**F06a**

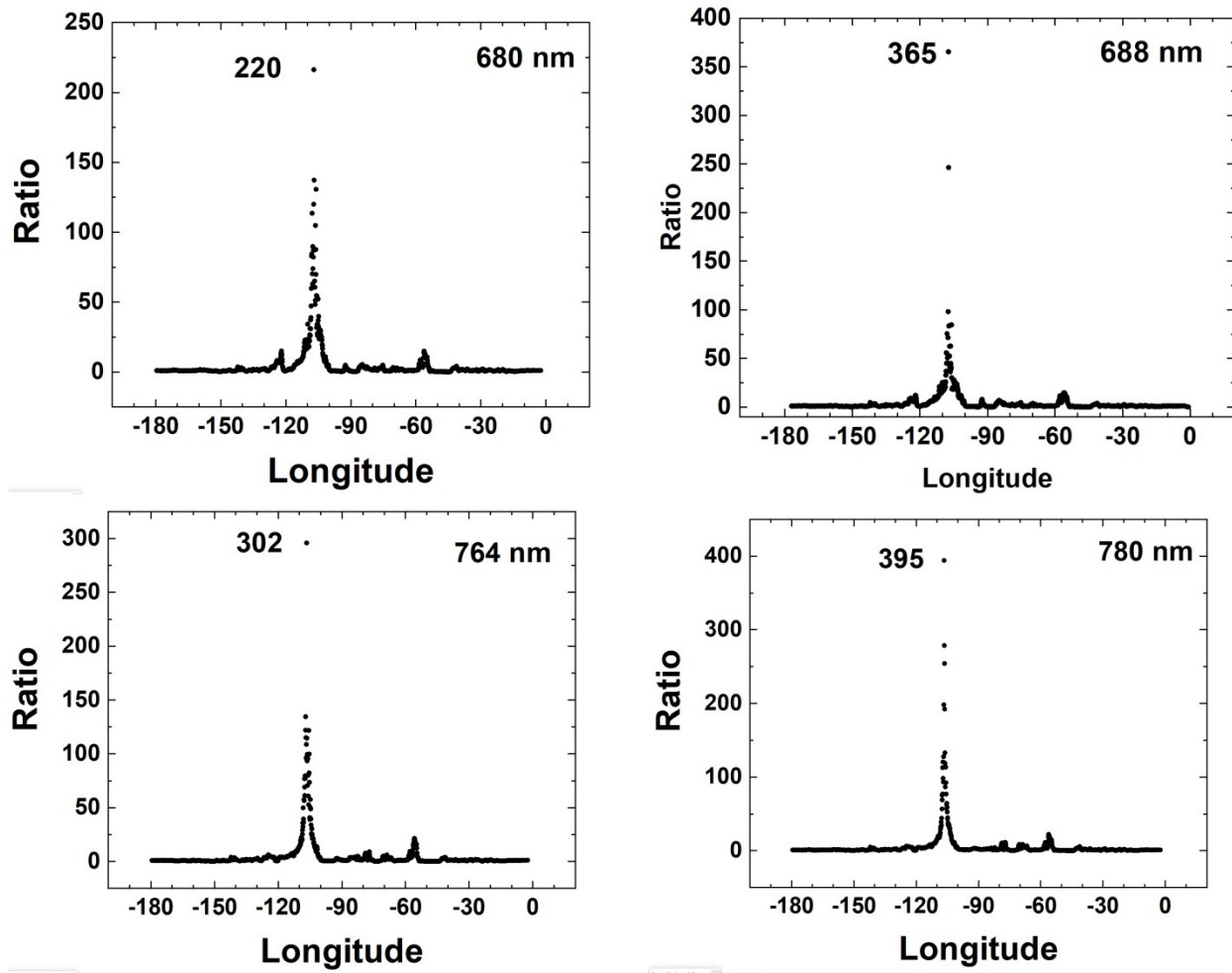

Fig. 6a. The ratio $R_{EN}(\lambda_i)$ = I(Aug20)/I(Aug21) at the time of the Eclipse in Casper Wyoming for wavelengths 317.5 to 780 nm. The channels 317.5 to 340 nm are affected by ozone absorption and the channels 688 and 764 nm are within the $O_2$ B and A absorption bands.



**F06a Continued**

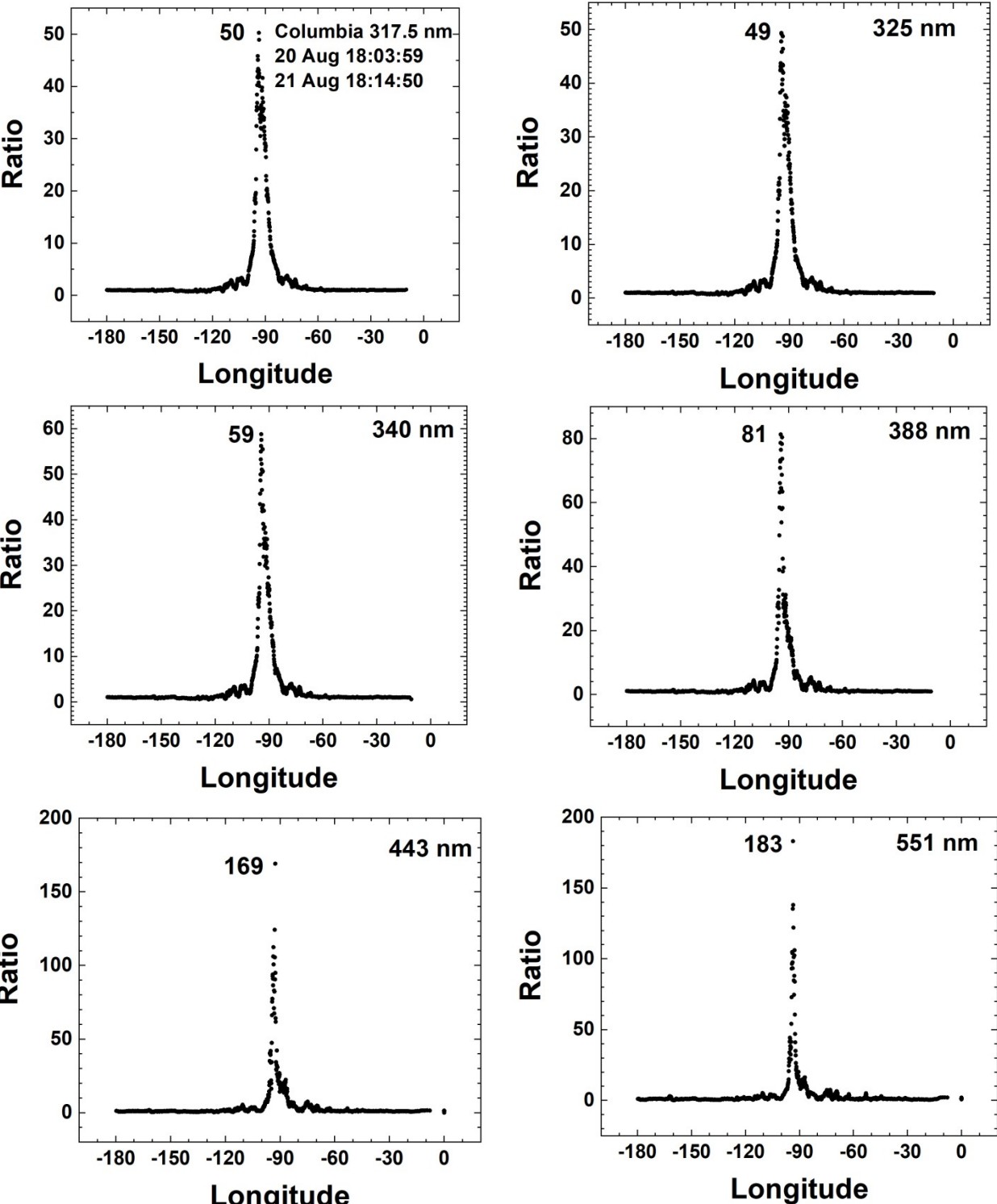


**F06b**

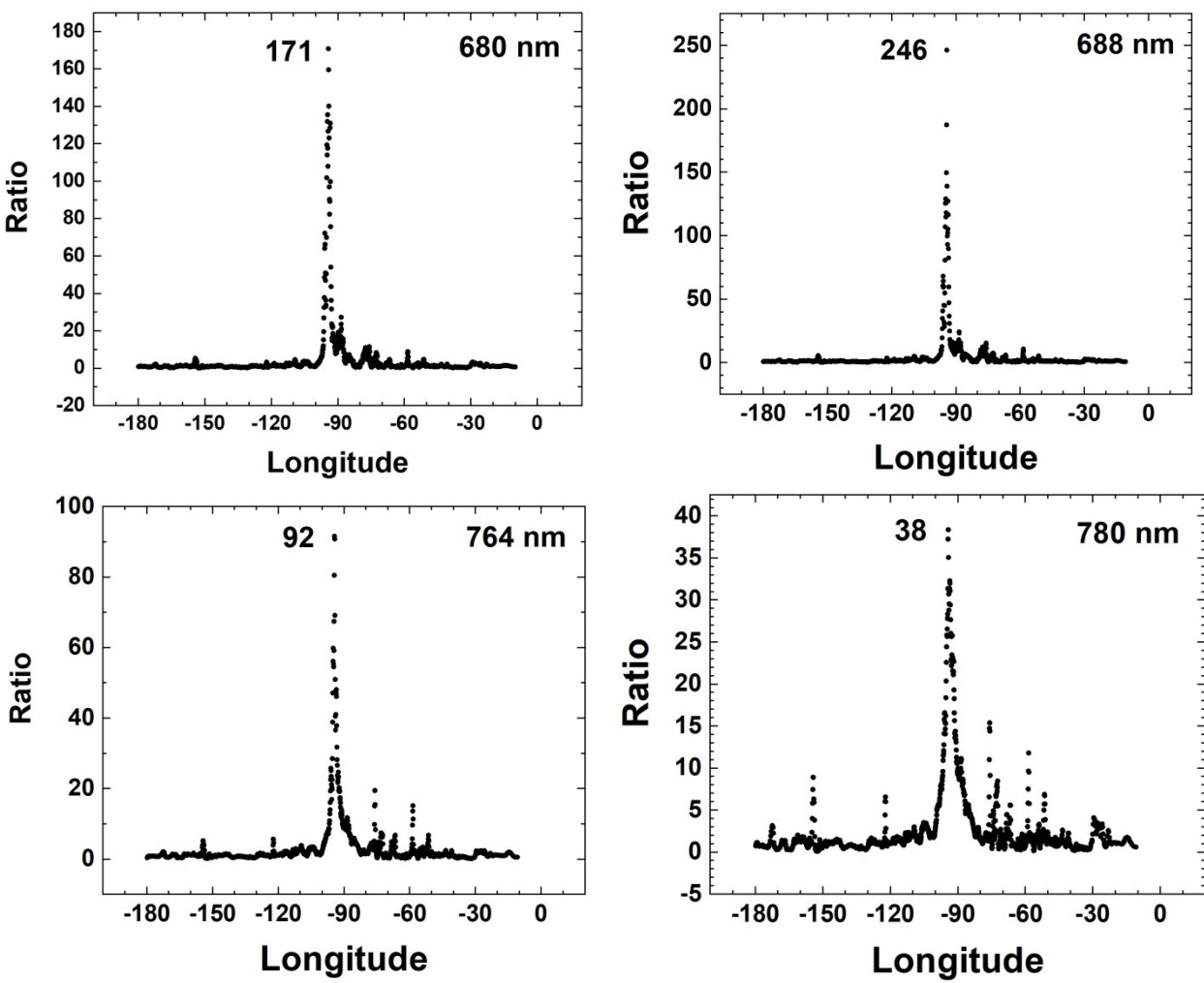

Fig. 6b. The ratio $R_{EN}(\lambda_i) = I(Aug20)/I(Aug21)$ at the time of the Eclipse in Columbia, Missouri for wavelengths 317.5 to 780 nm. The channels 317.5 to 340 nm are affected by ozone absorption and the channels 688 and 764 nm are within the $O_2$ B and A absorption bands.




**F06b  Continued**

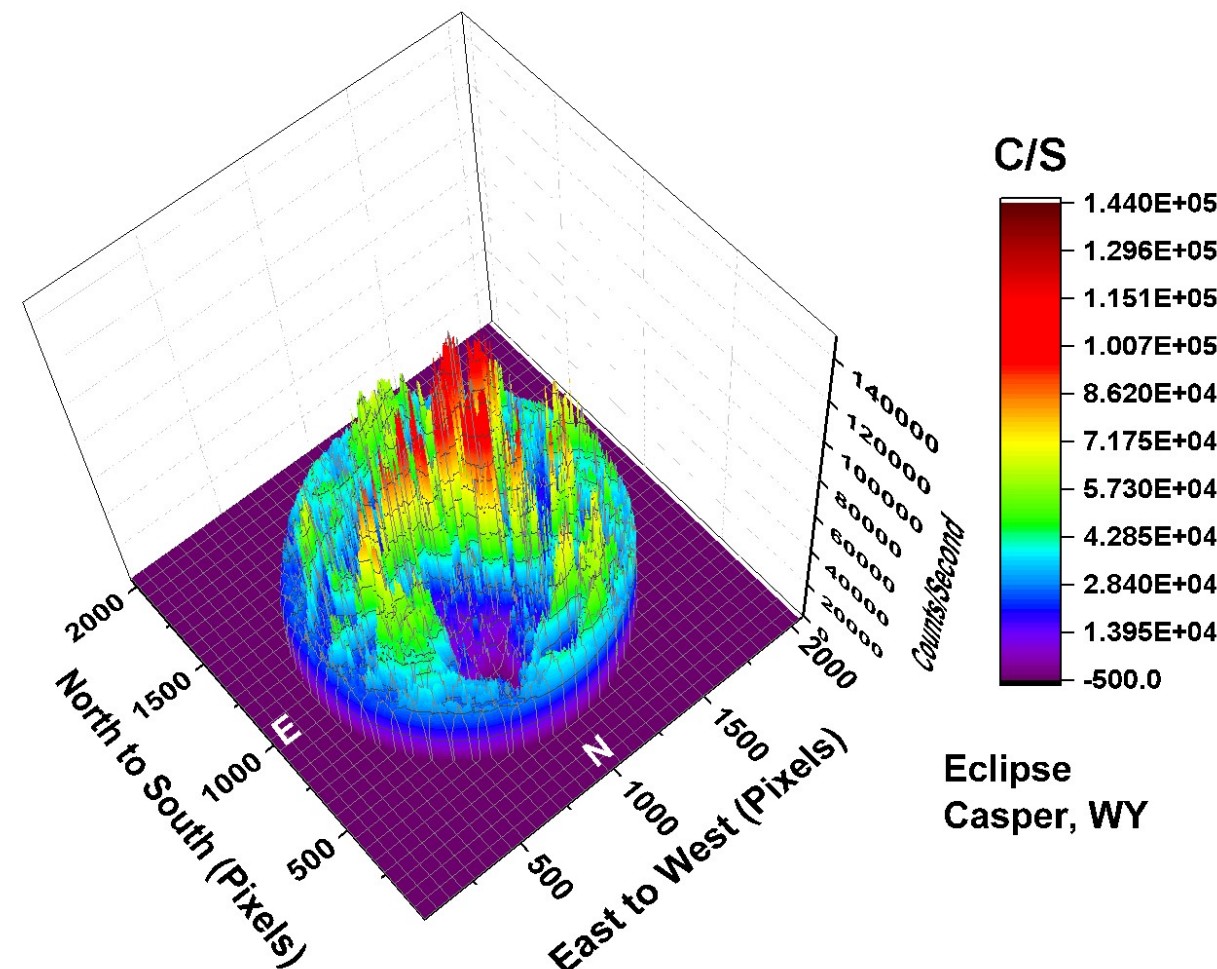

Fig. 7 The C/s observed by EPIC for the 443 nm channel corresponding to the color image shown in Fig. 1. In the data file, the word infinity has been replaced by the number zero. In this image there are approximately $2.59 \times 10^6$ illuminated pixels out of $2048^2 = 4.194304 \times 10^6$ pixels (61.8 %).



**F07**


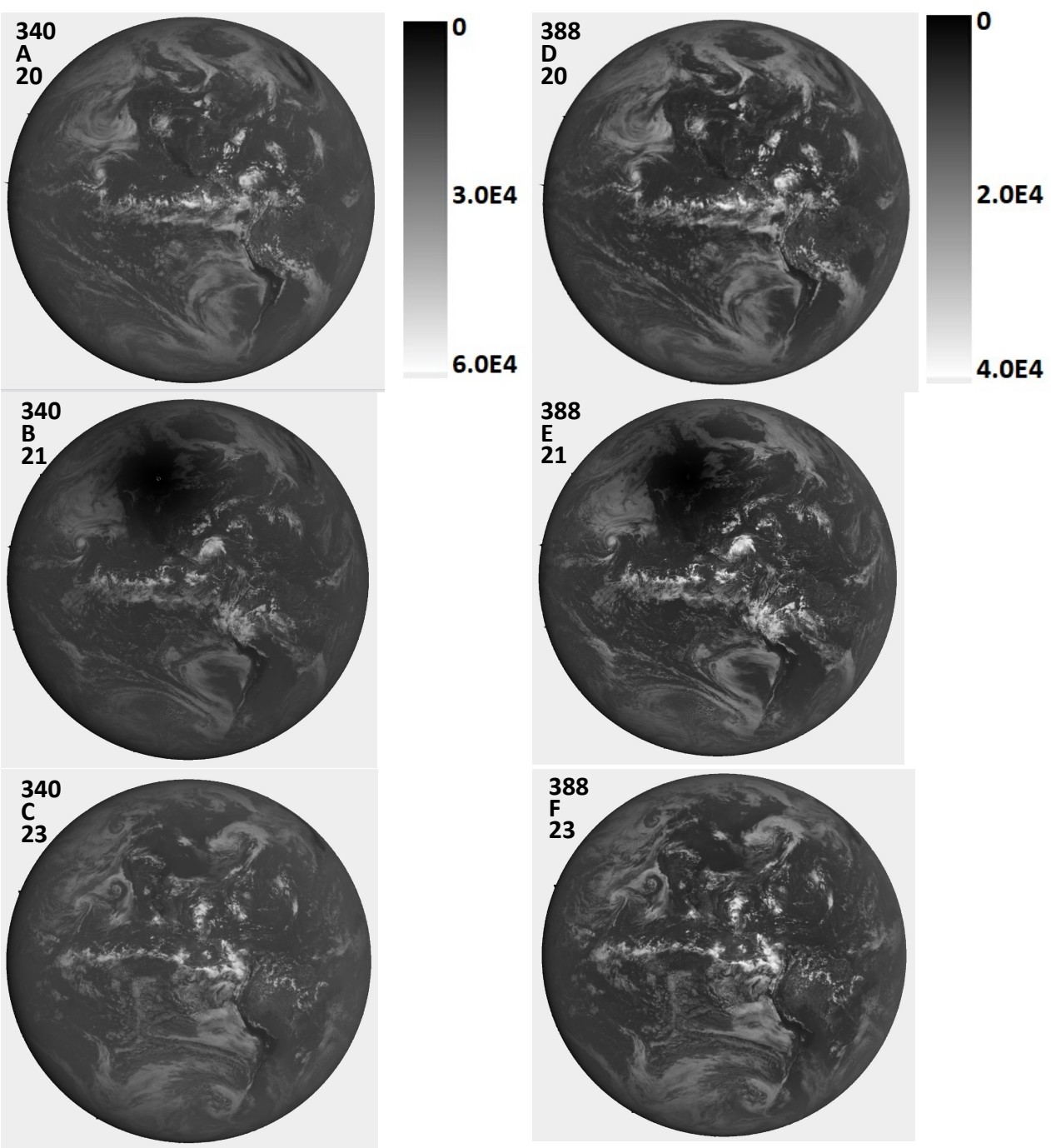

Figure 8a Image in C/s for 340 and 388 nm for 20 Aug.(A+D), 21 Aug. (B+E), and 23 Aug. (C+F). The scale applies to the specific wavelength. North is up.


**F08a**

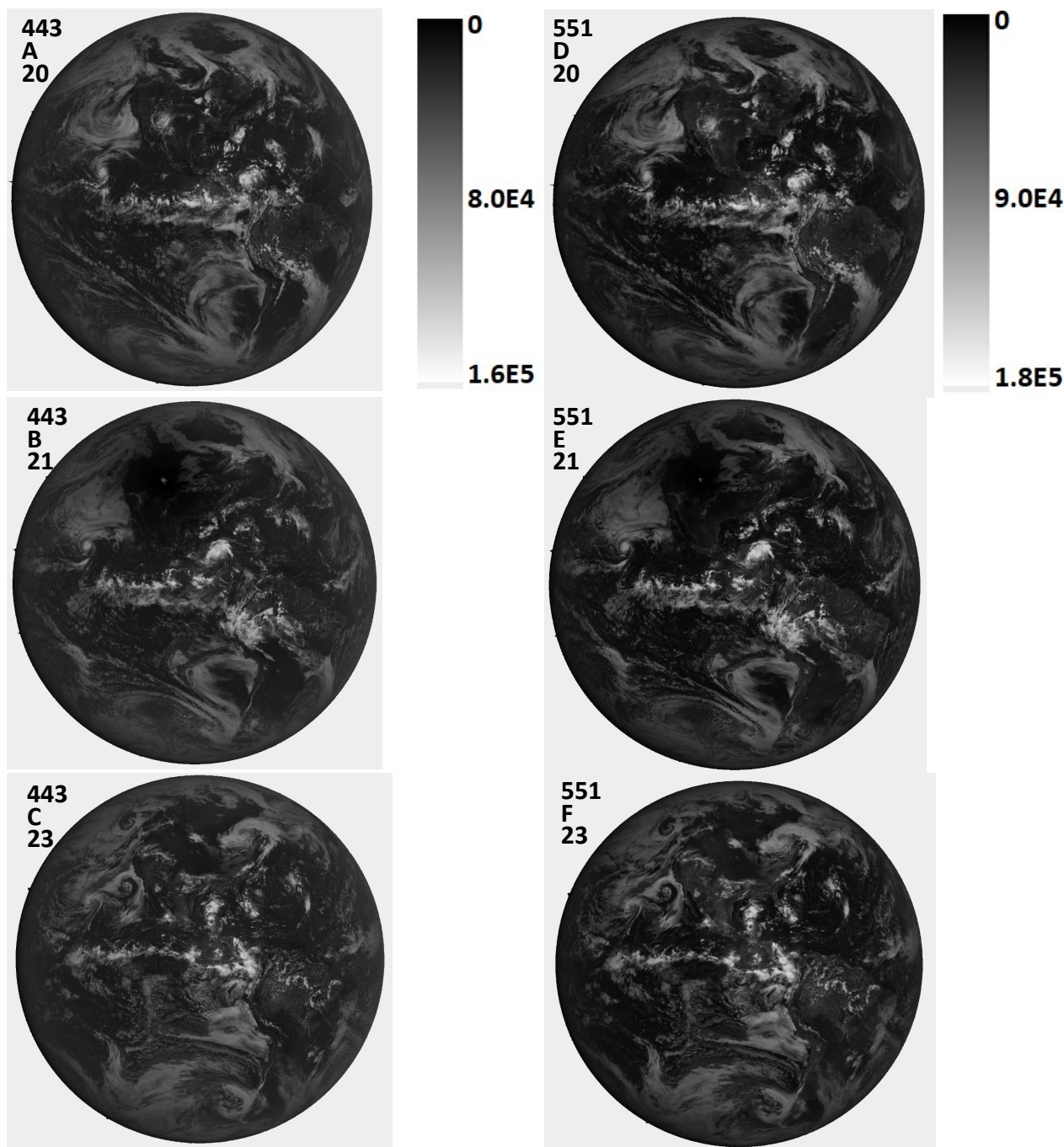

Figure 8b Image in C/s for 443 and 551 nm for 20 Aug.(A+D), 21 Aug. (B+E), and 23 Aug. (C+F). The scale applies to the specific wavelength. North is up.



**F08b**

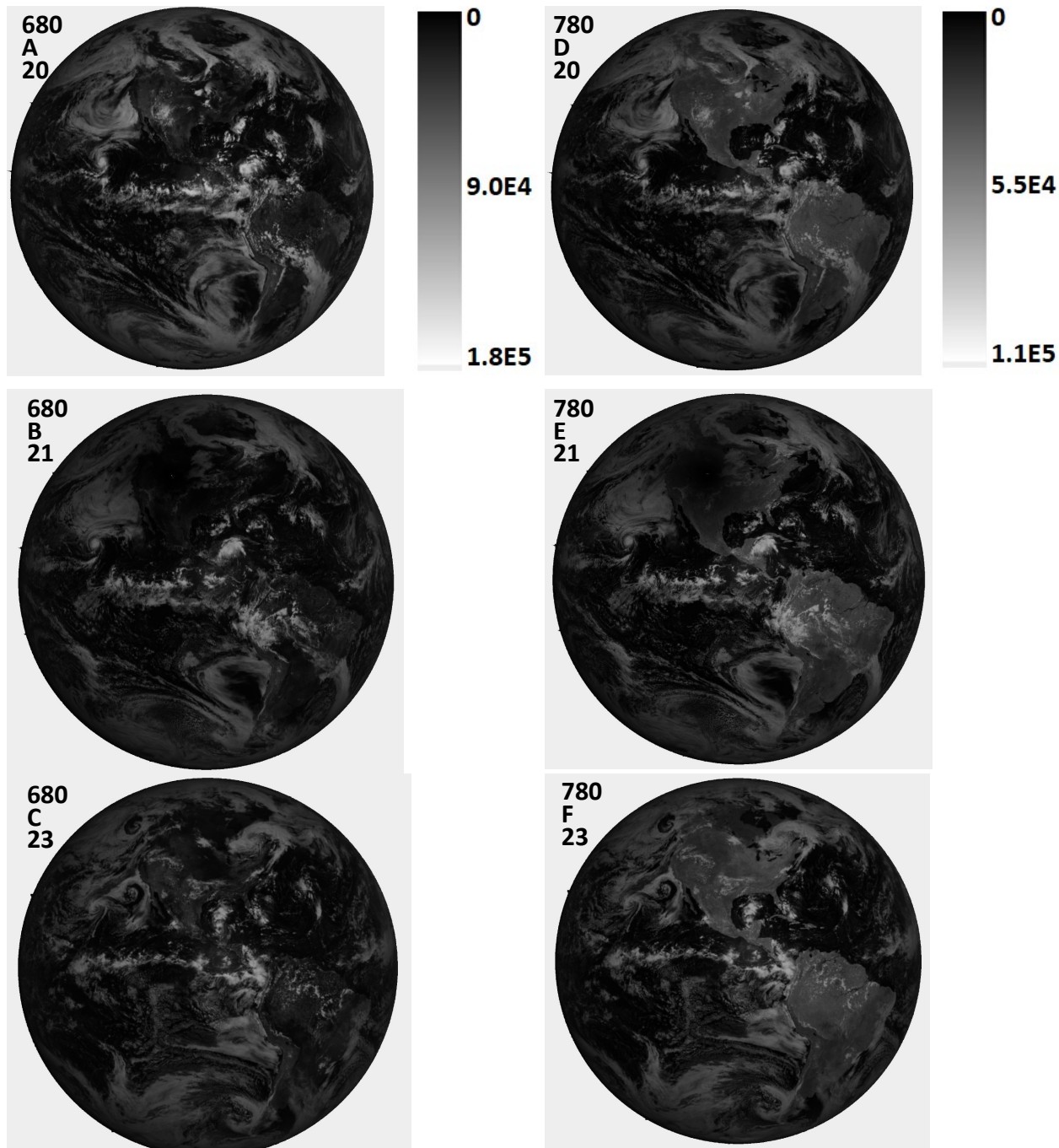

Figure 8c Image in C/s for 680 and 780 nm for 20 Aug.(A+D), 21 Aug. (B+E), and 23 Aug. (C+F). The scale applies to the specific wavelength. North is up.



**F08c**


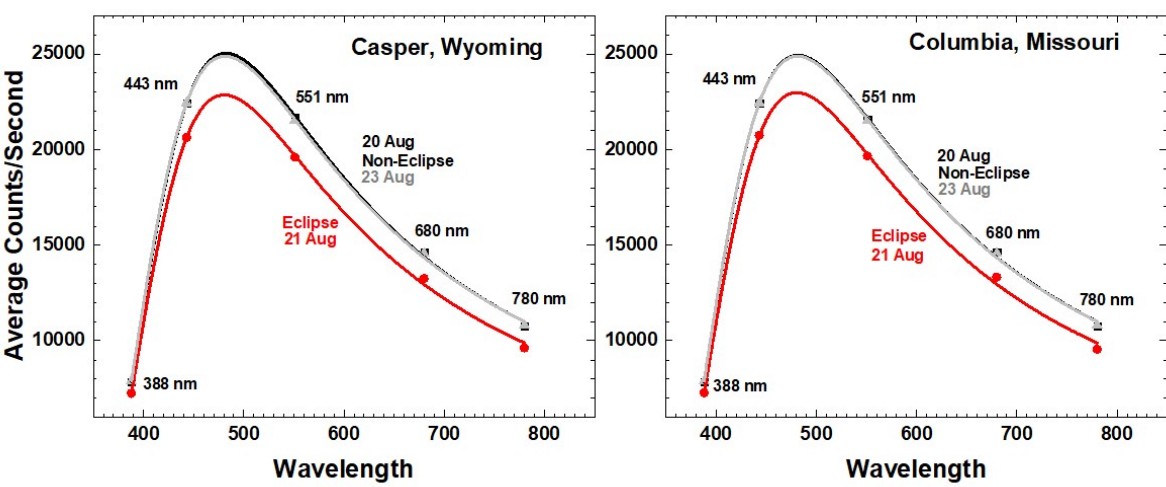

Fig. 9 Average reflected light in C/s for eclipse (21 Aug. red) and non-eclipse (20 Aug. and 23 Aug. (black and grey) days from Table 3 and Eqn. 1 for Casper and Columbia. The locations of the maxima are from curve fitting to the discrete wavelength measurements.



**F09**


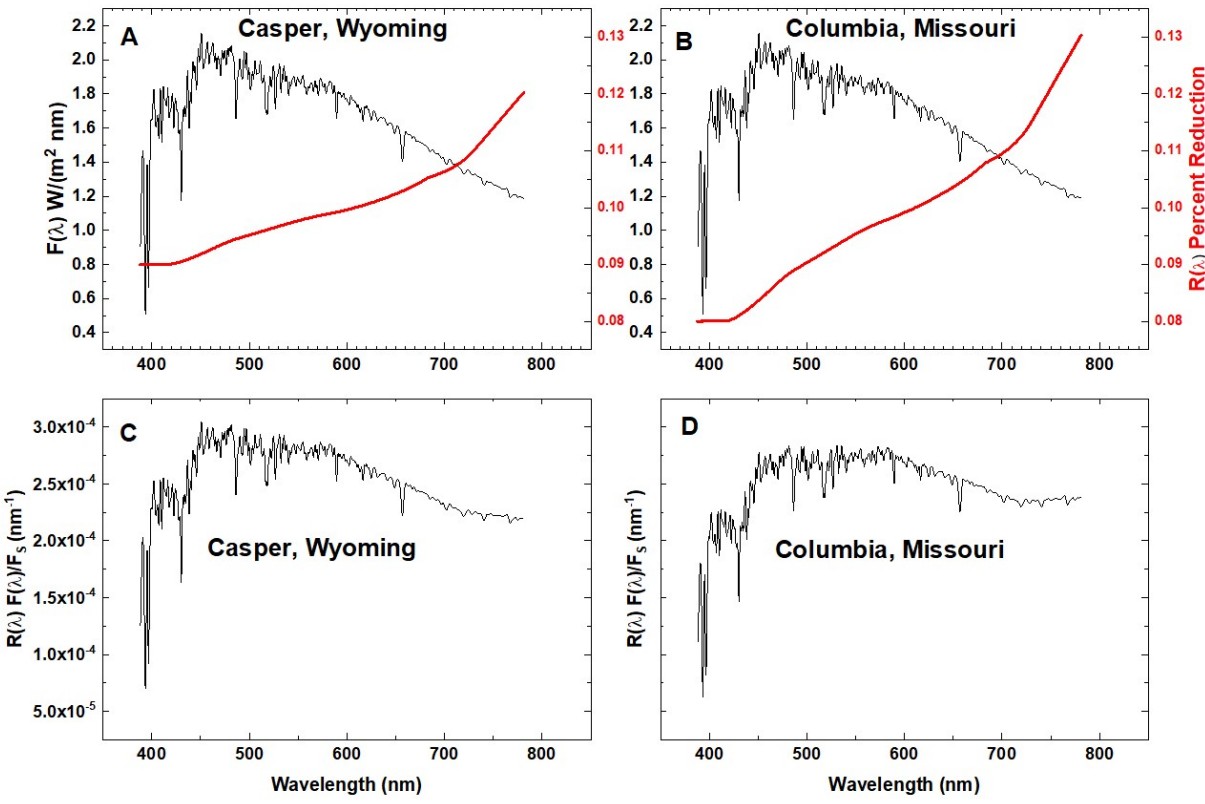

Fig. 10 Solar Irradiance at 1 AU F($\lambda$) Watts/(m$^2$ nm) (Mayer and Kylling, 2005) and the eclipse reduction function R($\lambda$) in percent for Casper, Wyoming (red curve in panel A) and Columbia, Missouri (red curve in panel B). Fractional reduction (nm$^{-1}$) in reflected solar irradiance in the direction of L-1 for Casper, Wyoming (panel C) and Columbia, Missouri (panel D)




**F10**

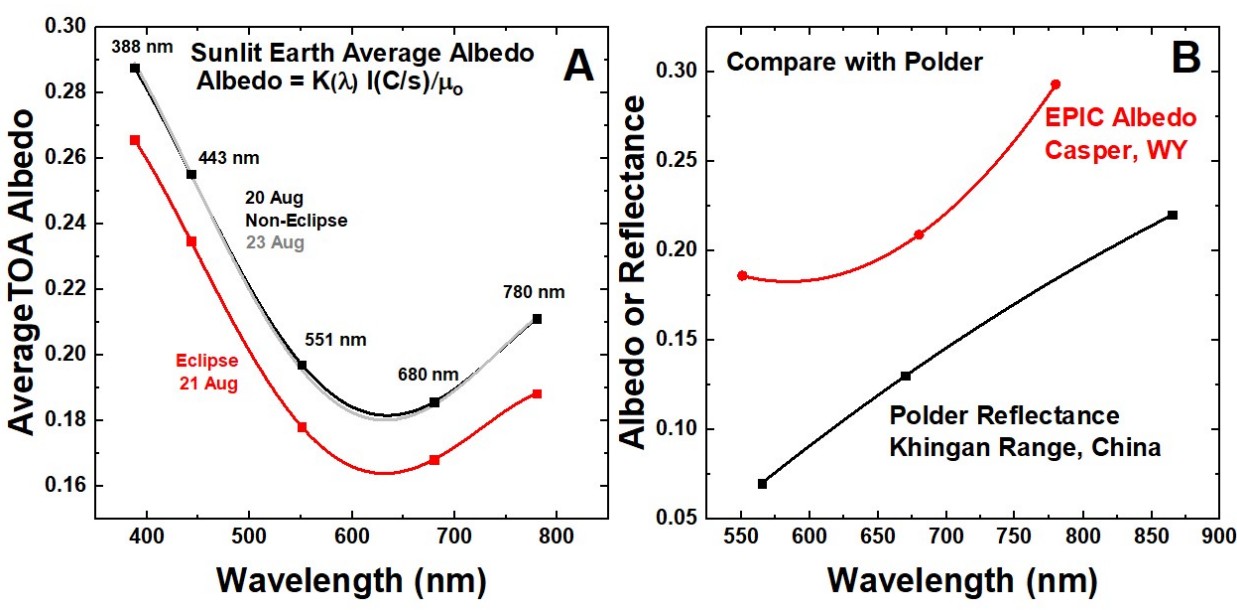

Fig. 11 A. The measured albedo at Casper Wyoming on 20 Aug (black curve) and 23 Aug (grey curve) compared to B the POLDER measured surface reflectance in the Khingan Range, China ( Maignan et al., 2004) corresponding to $8^{O}$ from overhead sun.


**F11**



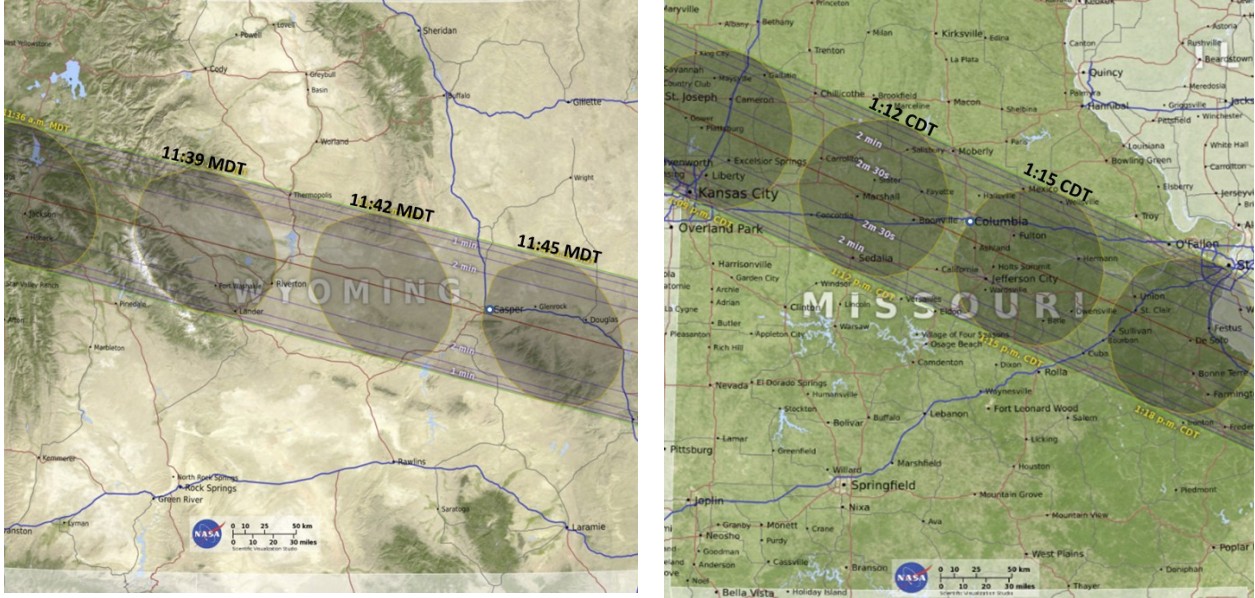

Fig. A1 The timing and shape of the Moon's shadow over Casper, Wyoming showing the relative location of Casper and Columbia (white circles) at 11:45 MDT (Mountain Daylight Time) and 1:15 CDT (Central Daylight Time). The shadow is moving at about 46 km/minute. (https://eclipse2017.nasa.gov/eclipse-maps).  The scale size with the NASA logo is 50 km.


**FA1**

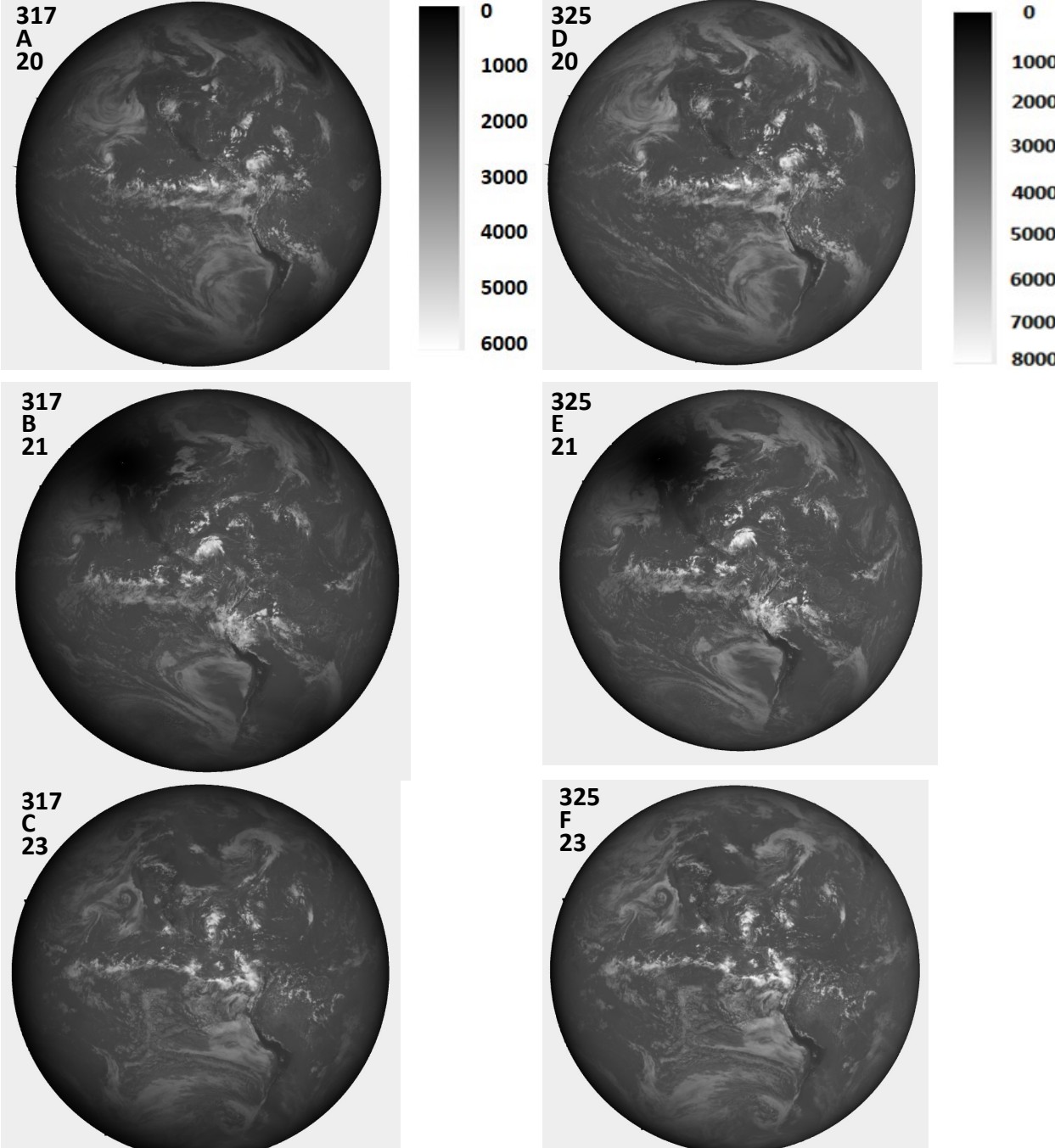

Fig. A2a Image in C/s for 317 and 340 nm for 20 Aug., 21 Aug. and 23 Aug. The scale applies to the specific wavelength. North is up.


**FA2**


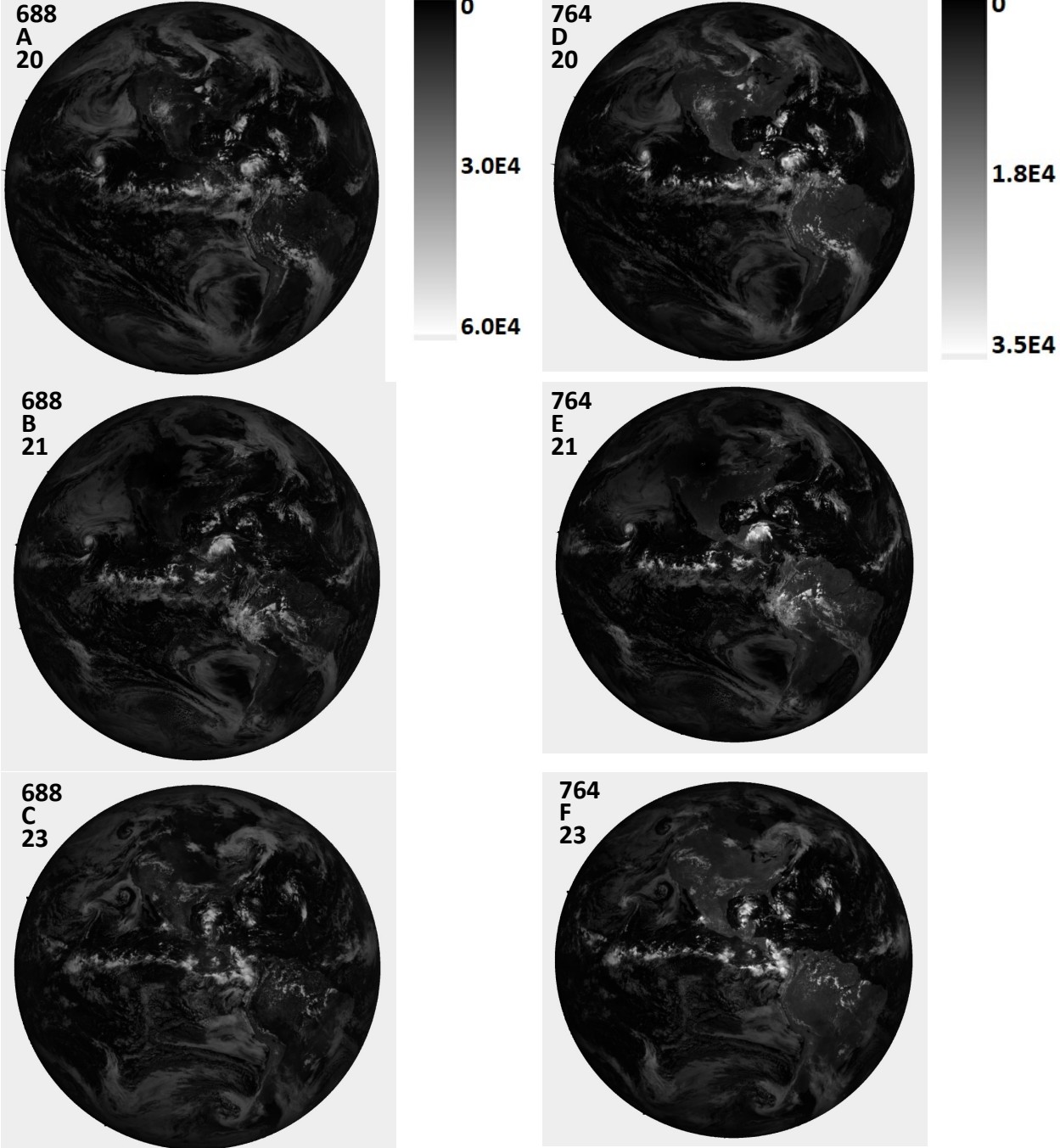

Fig. A2b Image in C/s for 688 and 764 nm for 20 Aug., 21 Aug. and 23 Aug. The scale applies to the specific wavelength. North is up.



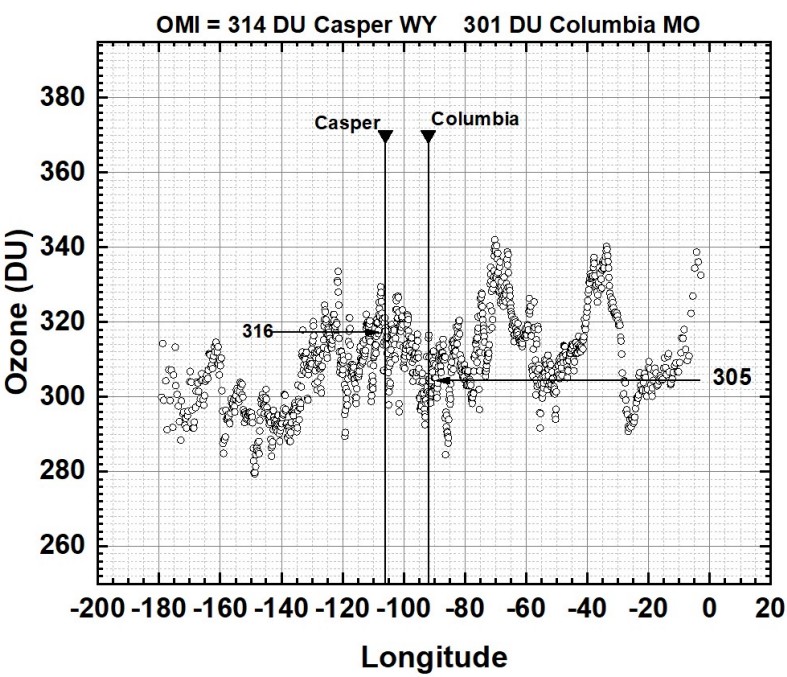

Fig. A3 EPIC measured ozone amounts from 20 August in the vicinity of Casper, WY and Columbia, MO.




**FA3**