# Peer review of "Reduction in sunlit Earth reflected radiance 317 to 780 nm during the eclipse of 21 August 2017"

_Atmospheric Measurement Techniques, 2017_

## Referee Comment (RC1) · Anonymous Referee #2 · 27 Mar 2018

The authors use DSCOVR/EPIC observations from 21st August 2017, which were taken in the Lagrange-1 point, to estimate the reduction of Earth reflected solar radiance during the solar eclipse. They compare images from the day before and the day after the eclipse and find a reduction of about 9.7% for a particular location (Casper, Wyoming). They find similar results for absorbing and non-absorbing channels, thus they conclude that the reflected energy is dominated by radiation reflected by clouds with a significant contribution of Rayleigh scattering at the shorter wavelengths. Further, for two locations, they have calculated the reduction of reflected radiance in the totality region and they found, that it highly depends on the cloud cover. When clouds are present, the reduction is much less than without clouds.

Since this is to my knowledge the first estimate of the reduction of the reflected radiance

from space, the result is interesting and it should be published.

However, before publication in AMT, the paper needs to be revised and some figures could be improved (see below).

General comments:

1. In the abstract and also later in the text (l. 237 ff, l. 366 ff), the observations are compared to modelling results by Emde and Mayer, 2007: "A previously published clear-sky model (Emde and Mayer, 2007) shows results for a nearly overhead eclipse had $R_{EN}$ (340nm)=1.7x10ˆ4 compared to the maximum measured non-averaged $R_{EN}$ (340) at Casper of 515±27 with optically thin clouds under similar geometrical conditions." Such a quantitative comparison is not possible, because the modelling result refers to the reduction of global irradiance (measured from the surface) in the center of the umbral shadow. This is a different quantity and the observation geometry is completely different, thus it is not surprising that the results do not agree to the DISCOVR observations. For a quantitative comparison, the 3D radiative transfer model needs a completely different setup, one has to model the reflected radiance for the specific observation geometry of DISCOVR (with a phase angle of about 172°). This should be possible using a Monte Carlo code like MYSTIC used by Emde and Mayer, 2007, but as said before, it requires a completely different setup. It can be mentioned in the text, that it would be interesting to model the observations with 3D RT models, but quantitative comparisons should be removed from abstract, text and summary.

2. Why is the global reduction of the reflectance (Section 3.2) only calculated for Casper and not for Columbia. It would be interesting to see, how much the reduction of global reflectance depends on the location of the clouds. Please include results for Columbia in Section 3.2.

3. Could the DISCOVR data, which is used for the study, added as supplementary data to the paper in addition to the provided links?

[Figure]

Specific comments:

l.1 "Sunlit" -> "sunlit"

l. 16: "A reduction of 9.7±1.7% in the radiance (387 to 781 nm) reflected from the Earth towards L1 was obtained..." -> Please clarify that this is the spectrally integrated global reflectance.

l. 25ff:"A previously published ..." -> remove this sentence from abstract (see above).

l.45 "earth" -> "Earth"

l. 55: "The totality region (umbra) is about 250 to 267 km in diameter, ..." -> in line 151 it is said that the totality shadow is 116 km wide over Caspar. How do these numbers match? In l. 151ff both axes of the umbral shadow should be provided for both locations (Caspar and Columbia). On which parameters does the size of the shadow depend apart from solar zenith angle? The distances Sun-Earth, Moon-Earth also determine the extent of the shadow.

l. 60ff: The overview paper of Gerasopoulos et al., 2008 is cited and it is said that it would include MODIS observations of the eclipse from 2006 over Europe. This is not correct, the paper includes a MODIS image from the same day (taken before eclipse) to show the cloud formation over Greece.

l.66ff: "A 3D Monte Carlo radiative transfer study (Emde and Mayer, 2007) was applied to the geometry for the nearly overhead total eclipse of 29 March 2006 (13:20 local time in Turkey), but without the effect of clouds included in the calculation. Successful modelling of an eclipse under realistic conditions is the first step to improved modelling ..." -> The modelling was realistic for the given observation over Greece, because the region was cloud-free. A comparison to observations showed an excellent agreement (see Kadzantzidis et al. 2007, https://www.atmos-chem-phys.net/7/5959/2007/). Of course, clouds need to be included when present. Please clarify this sentence.

l. 71ff: "The observations from the DSCOVR satellite are part of a larger project that

combines simultaneously obtained satellite and ground-based measurements using a pyranometer (Ji and Tsay, 2000) and the Pandora Spectrometer Instrument (Herman et al., 2009) at both sites." -> Is this data available?

l. 75: "This study presents the only synoptic satellite data of the sunlit Earth ever obtained during an eclipse ..." -> What about images from geostationary satellites? I remember movies of MSG images of the eclipse from March 2006.

l.119: "To reduce the volume of data, all measurements, except those from the 443 nm channel, were averaged onboard DSCOVR to 1024 x 1024 pixels." -> Why is the 443nm channel treated differently?

l. 167: "340 nm, with strong Rayleigh scattering effects (haze)" -> haze (aerosol) scattering is not the same as Rayleigh (molecular) scattering.

Fig.3: These images are very nice. I would prefer north up as usual, even if inconsistent with Fig. 7.

l. 173 "3.1 Comparison of Eclipse and Non-Eclipse Days for Caspar, WY and Columbia, MO "-> Use a more specific title, what is compared?

Fig. 5: "Middle"-> "Bottom"

Fig. 6a,b: These figures all look quite similar. Why are all channels shown for Caspar, WY and only one for Columbia (as lower left plot in Fig 6a). This arrangement is confusing. Suggestion: Use 3 representative channels and show the results for Caspar, WY on the left and the corresponding results for Columbia on the right. Maximum values for all channels and for both locations should be included in Table 2.

Table 2: Please include also the <R_EN> values for Columbia in the table.

l. 234ff: "A detailed radiative transfer study for realistic conditions is made feasible by using EPIC's simultaneous estimates of cloud reflectivity and transmission, cloud height, aerosol amounts, and ozone amounts." -> It there a data product including

these estimates. If yes, please provide reference.

l. 237ff: see "general comment 1"

Fig.8 a,b,c, A2: I think that not all of these figures are needed. Fig. 3 nicely shows, how the shadow and the Earth looks in various channels and Fig. 4 shows the synoptical conditions during the eclipse and the day before and after the eclipse. I suggest to put most of these figures in the appendix. l. 285: "While the figures are similar from wavelength to wavelength, there are differences in the depth of the eclipse totality and the reflectivities of the surrounding clouds." With the chosen grey-scale colormaps these difference are not visible. I suggest to use a colormap similar to the one in Fig.7 to visualize the differences in the depth of the eclipse totality.

Table 3: Please include this table also for Columbia.

l. 325: "This means that EPiC is observing close to "hotspot" conditions where the backscatter amount increases with increasing wavelength (Maignan et al., 2004). At 551 and 680 nm the hotspot effect is smaller than at 780 nm." -> This is not obvious from Table 3, integrated counts are much larger for 551nm than for 780nm. Please explain.

l. 330: "The solar spectrum used is a combination of data named atlas_plus_modtran (Mayer and Kylling, 2005)." -> Mayer and Kylling is the reference for the libRadtran software package, from which the solar irradiance data is taken. Please rewrite sentence more clearly.

l. 363: "A previously published clear-sky model result for a nearly overhead eclipse ratios and an ocean surface albedo of 0.06 (Emde and Mayer, 2007) had $R\_EN$ (340nm)=$1.7 \times 10^4$ compared to the measured non-averaged $R\_EN$ (340) at Casper of $515 \pm 27$ with optically thin clouds under similar geometrical conditions." -> these results can not be compared (see general comments). Not $R\_EN$ has been modelled by Emde and Mayer, the value refers to global irradiance at the surface!

---

## Author Comment (AC1) · 8 Apr 2018

*Based on your comments, I have made a lot of changes to the text and calculations in the paper. All changes are indicated in* ==GREEN==. *The altered version is attached at the end.*

The authors use DSCOVR/EPIC observations from 21st August 2017, which were taken in the Lagrange-1 point, to estimate the reduction of Earth reflected solar radiance during the solar eclipse. They compare images from the day before and the day after the eclipse and find a reduction of about 9.7% for a particular location (Casper, Wyoming). They find similar results for absorbing and non-absorbing channels, thus they conclude that the reflected energy is dominated by radiation reflected by clouds with a significant contribution of Rayleigh scattering at the shorter wavelengths. Further, for two locations, they have calculated the reduction of reflected radiance in the totality region and they found, that it highly depends on the cloud cover. When clouds are present, the reduction is much less than without clouds.

Since this is to my knowledge the first estimate of the reduction of the reflected radiance from space, the result is interesting and it should be published.

However, before publication in AMT, the paper needs to be revised and some figures could be improved (see below).

General comments:

1. In the abstract and also later in the text (l. 237 ff, l. 366 ff), the observations are compared to modelling results by Emde and Mayer, 2007: "A previously published clear-sky model (Emde and Mayer, 2007) shows results for a nearly overhead eclipse had R EN (340nm)=1.7x10ˆ4 compared to the maximum measured non-averaged R EN (340) at Casper of 515±27 with optically thin clouds under similar geometrical conditions." Such a quantitative comparison is not possible, because the modelling result refers to the reduction of global irradiance (measured from the surface) in the center of the umbral shadow. This is a different quantity and the observation geometry is completely different, thus it is not surprising that the results do not agree to the DISCOVR observations. For a quantitative comparison, the 3D radiative transfer model needs a completely different setup, one has to model the reflected radiance for the specific observation geometry of DISCOVR (with a phase angle of about 172˚). This should be possible using a Monte Carlo code like MYSTIC used by Emde and Mayer, 2007, but as said before, it requires a completely different setup. It can be mentioned in the text, that it would be interesting to model the observations with 3D RT models, but quantitative comparisons should be removed from abstract, text and summary.

2. Why is the global reduction of the reflectance (Section 3.2) only calculated for Casper and not for Columbia. It would be interesting to see, how much the reduction of global reflectance depends on the location of the clouds. Please include results for Columbia in Section 3.2.

**I have added the Columbia calculation. It is almost the same as the Casper calculation, since the earth's cloud cover did not change much in the short time interval between the eclipse at Casper and Columbia.**

3. Could the DISCOVR data, which is used for the study, added as supplementary data to the paper in addition to the provided links?

**The data are available freely from** https://eosweb.larc.nasa.gov/project/dscovr/dscovr_epic_l1b, **but I will try to upload the data files with some extra documentation.**

Specific comments:

l.1 "Sunlit" -> "sunlit"  **OK**

l. 16: "A reduction of 9.7±1.7% in the radiance (387 to 781 nm) reflected from the Earth towards L1 was obtained..." -> Please clarify that this is the spectrally integrated global reflectance.

**OK**

l. 25ff:"A previously published ..." -> remove this sentence from abstract (see above). l.45

"earth" -> "Earth"  **OK**

l. 55: "The totality region (umbra) is about 250 to 267 km in diameter, ..." -> in line 151 it is said that the totality shadow is 116 km wide over Caspar. How do these numbers match?
Fixed. My mistake
In l. 151ff both axes of the umbral shadow should be provided for both locations (Caspar and Columbia). On which parameters does the size of the shadow depend apart from solar zenith angle? The distances Sun-Earth, Moon-Earth also determine the extent of the shadow.

**I changed the sentence to read: "The totality region (umbra) is an oval of about 110 -120 km in size near local noon at Casper, Wyoming and Columbia, Missouri, but will change size and shape as a function of local solar zenith angle (https://eclipse2017.nasa.gov/eclipse-maps)."**

**The umbral size does depend a little on the Sun-Earth distance, but for this particular eclipse, the size just depends on the local solar zenith angle.**

l. 60ff: The overview paper of Gerasopoulos et al., 2008 is cited and it is said that it would include MODIS observations of the eclipse from 2006 over Europe. This is not correct, the paper includes a MODIS image from the same day (taken before eclipse) to show the cloud formation over Greece.

That is correct. I changed the sentence to read: ".....observations of cloud cover before totality (Gerasopoulos et al., 2008)...."

l.66ff: "A 3D Monte Carlo radiative transfer study (Emde and Mayer, 2007) was applied to the geometry for the nearly overhead total eclipse of 29 March 2006 (13:20 local time in Turkey), but without the effect of clouds included in the calculation. Successful modelling of an eclipse under realistic conditions is the first step to improved modelling ..." -> The modelling was realistic for the given observation over Greece, because the region was cloud-free. A comparison to observations showed an excellent agreement (see Kadzantzidis et al. 2007, https://www.atmos-chem-phys.net/7/5959/2007/). Of course, clouds need to be included when present. Please clarify this sentence.

**Thank you. I missed that reference.  I added the sentence: "The 3D model showed good agreement with fairly cloud-free (few cumulus, 1-2 octas, and scattered cirrus, 3-4 octas) measurements at 380 nm of the ratio of global irradiance starting 5 minutes before totality to that during totality (Kazantzidis et al., 2007)."**

l. 71ff: "The observations from the DSCOVR satellite are part of a larger project that combines simultaneously obtained satellite and ground-based measurements using a pyranometer (Ji and Tsay, 2000) and the Pandora Spectrometer Instrument (Herman et al., 2009) at both sites." -> Is this data available?

**The data are available, but is the subject of another paper by Guoyong Wen, the main investigator for the project.**

l. 75: "This study presents the only synoptic satellite data of the sunlit Earth ever obtained during an eclipse ..." -> What about images from geostationary satellites? I remember movies of MSG images of the eclipse from March 2006.

**You are correct about images, but not spectral calibrated data. I added the word spectral.**

**"This study presents the only calibrated spectral synoptic satellite data of the sunlit Earth ever obtained during an eclipse".**

l.119: "To reduce the volume of data, all measurements, except those from the 443 nm channel, were averaged onboard DSCOVR to 1024 x 1024 pixels." -> Why is the 443nm channel treated differently?

**The EPIC project wanted at least one high resolution channel to help with geolocation of the images and to enhance the resolution of the RGB color pictures.**

l. 167: "340 nm, with strong Rayleigh scattering effects (haze)" -> haze (aerosol) scattering is not the same as Rayleigh (molecular) scattering.

**I should not have used the word haze – I was using it subjectively. It has been removed**

Fig.3: These images are very nice. I would prefer north up as usual, even if inconsistent with Fig. 7. **I usually prefer north up, but went for consistency instead. If it is really important, I can rotate the figures with north up.**

l. 173 "3.1 Comparison of Eclipse and Non-Eclipse Days for Caspar, WY and Columbia, MO "-> Use a more specific title, what is compared?

**It's a bit long, but: "Comparison of EPIC Observations of Eclipse Totality (21 Aug) with Non-Eclipse**

**Days (20 and 23 Aug) for Casper, WY and Columbia, MO.**

Fig. 5: "Middle"-> "Bottom"  **Fixed. The original figure had an addition row**

Fig. 6a,b: These figures all look quite similar. Why are all channels shown for Caspar, WY and only one for Columbia (as lower left plot in Fig 6a). This arrangement is confusing. Suggestion: Use 3 representative channels and show the results for Caspar, WY on the left and the corresponding results for Columbia on the right. Maximum values for all channels and for both locations should be included in Table 2.

**I calculated the results for all 10 channels for both Casper and Columbia, but did it differently than in the original document. In the original document I used a Lowess running average of 12 data points to produce a smooth curve. I forgot to add that to the figure caption, but it was in the text. In the present version, I am showing the original data points with no spatial averaging. The maximum values are shown in the figures and Table 2.**

Table 2: Please include also the <R_EN> values for Columbia in the table. **OK**

l. 234ff: "A detailed radiative transfer study for realistic conditions is made feasible by using EPIC's simultaneous estimates of cloud reflectivity and transmission, cloud height, aerosol amounts, and ozone amounts." -> It there a data product including these estimates. If yes, please provide reference.

**Ozone, cloud reflectivity, and cloud transmission:**

**Herman, Jay, Liang Huang, Richard McPeters, Jerry Ziemke, Alexander Cede, and Karin Blank, Synoptic ozone, cloud reflectivity, and erythemal irradiance from sunrise to sunset for the whole earth as viewed by the DSCOVR spacecraft from the earth–sun Lagrange 1 orbit, 2017, Atmos. Meas. Tech., 10, 1–18, https://doi.org/10.5194/amt-10-1-2017.**

**OK  I added an ozone figure in the appendix.**

**I am not sure if the aerosol paper has been published yet. The lead author will be Omar Torres.**

l. 237ff: see "general comment 1"

Fig.8 a,b,c, A2: I think that not all of these figures are needed. Fig. 3 nicely shows, how the shadow and the Earth looks in various channels and Fig. 4 shows the synoptical conditions during the eclipse and the day before and after the eclipse. I suggest to put most of these figures in the appendix.

**The figures for 340 and 388 nm have been moved to the appendix. The main conclusion of this paper is based on a ratio of data from 20 Aug and 21 Aug. Most of the contribution to reflected radiances comes from the 443, 551, 680, and 780 nm channels.**

l. 285: "While the figures are similar from wavelength to wavelength, there are differences in the depth of the eclipse totality and the reflectivities of the surrounding clouds." With the chosen grey-scale colormaps these difference are not visible. I suggest to use a colormap similar to the one in Fig.7 to visualize the differences in the depth of the eclipse totality.

**I was asked by the editor after the original submission to show all of the wavelengths in a form where you could see the land areas. I was unable to do this in color contour plots, but could do it in greyscale.**

Table 3: Please include this table also for Columbia. **OK**

l. 325: "This means that EPiC is observing close to "hotspot" conditions where the backscatter amount increases with increasing wavelength (Maignan et al., 2004). At 551 and 680 nm the hotspot effect is smaller than at 780 nm." -> This is not obvious from Table 3, integrated counts are much larger for 551nm than for 780nm. Please explain.

**The results appear to be in contrast to measurements from Polder (Maignan et al., 2004), which show increasing reflectivity with increasing wavelength over northern China. There are two main differences in the EPIC data (Table 3). First, the data are measured C/s($\lambda$) that are basically raw radiance measurements proportional to the solar irradiance wavelength dependence (decreases with increasing wavelength after about 550 nm). Second, the Polder measurements are surface reflectances over land, where reflectivity of the surface increases at longer wavelengths. EPIC is measuring top of the atmosphere radiances (in C/s) that can be converted to top of the atmosphere reflectances using the calibration coefficients. In addition, the numbers in Table 3 are for a mixture of land and oceans.**

**The TOA eclipse measurements made by EPIC are near the backscatter direction (172[O]) for the incident solar irradiance over nearly cloud-free scenes. For land surfaces, such as the observations made at Casper and Columbia, measurements from the POLDER satellite over China show that the backscatter amount from the land surface increases with increasing wavelength (Maignan et al., 2004). For EPIC data over land that are comparable to the POLDER measurements, the C/s data should be converted to reflectance. When this is done, the wavelength dependence of EPIC is similar to POLDER even though there is no atmospheric correction and there is some cloud cover.**

**To directly answer your question I plotted the POLDER reflectance and the EPIC albedo for 172[O] backscatter.**

[Figure]

l. 330: "The solar spectrum used is a combination of data named atlas_plus_modtran (Mayer and Kylling, 2005)." -> Mayer and Kylling is the reference for the libRadtran software package, from which the solar irradiance data is taken. Please rewrite sentence more clearly.

l. 363: "A previously published clear-sky model result for a nearly overhead eclipse ratios and an ocean surface albedo of 0.06 (Emde and Mayer, 2007) had R_EN (340nm)=1.7x10ˆ4 compared to the measured non-averaged R_EN (340) at Casper of 515±27 with optically thin clouds under similar geometrical conditions." -> these results can not be compared (see general comments). Not R_EN has been modelled by Emde and Mayer, the value refers to global irradiance at the surface!

**The sentence has been modified to: "A previously published downward global surface radiation 
[revised manuscript text omitted]

---

## Referee Comment (RC2) · Anonymous Referee #1 · 23 Apr 2018

The paper has considerably improved.

Most of the my comments and recommendations have been already mentioned by reviewer #1. From my point of view the Maignan et al., 2004 study has to be reported in the introduction sections and the "differences" observed with this study have to be reported in more detail in the section that currently this work is mentioned.

I think this is an interesting and unique work that should be published in AMT.

---

## Author Comment (AC2) · 11 May 2018

The paper has considerably improved. Most of the my comments and recommendations have been already mentioned by reviewer #1. From my point of view the Maignan et al., 2004 study has to be reported in the introduction sections and the "differences" observed with this study have to be reported in more detail in the section that currently this work is mentioned.

I think this is an interesting and unique work that should be published in AMT.

I have added a new figure and text describing the comparison between POLDER and EPIC related to the Maignan et al. 2004 paper. I have also added a mention of the Maignan et al. paper in the introduction.

Measured backscattered radiances of the entire sunlit Earth were obtained during the 21 August 2017 eclipse from EPIC (Earth Polychromatic Imaging Camera) on the DSCOVR (Deep Space Climate Observatory) satellite. EPIC obtains synoptic observations of the Earth from an orbit around the $L_1$ point (Lagrange 1) 1.5 million km from Earth (Herman et al., 2018). EPIC top of the atmosphere TOA albedo measurements, made at a backscatter angle of $172^O$, are in the enhanced reflectivity regime (hotspot angles). EPIC non-eclipse day TOA albedos are compared to POLDER surface reflectivity measurements at $8^O$ (Maignan et al., 2004). This study focuses on data from two selected locations during the 21 August

354        The TOA eclipse measurements made by EPIC are near the hotspot backscatter direction (172°)
355   for the incident solar irradiance over nearly cloud-free scenes. For land surfaces, such as the
356   observations made at Casper and Columbia, measurements from the POLDER satellite over Khingan
357   Range, China (117.55°E to131.56°E, 45.68°N to 53.56°N) show that the backscatter amount from the
358   land surface increases with increasing wavelength (Maignan et al., 2004). For EPIC albedo data over
359   grassland that are comparable to the POLDER measurements, the C/s data in Fig. 9 can be converted to
360   TOA albedo.  When this is done (Fig. 10), the wavelength dependence of the EPIC TOA albedo (551, 680,
361   and 780 nm) is similar to POLDER surface reflectance at 8° even though there is no EPIC atmospheric
362   correction and there is some light cloud cover. The average TOA albedo from EPIC was almost the same
363   on 20 Aug. as on 23 Aug. (Fig.10A).
364

[Figure]

Fig. 10 A. The measured albedo at Casper Wyoming on 20 Aug (black curve) and 23 Aug (grey curve) compared to B the POLDER measured surface reflectance in the Khingan Range, China ( Maignan et al., 2004) corresponding to 8° from overhead sun.

365

366        The shape and magnitude differences are partially caused by the atmospheric component of the
367   albedo that includes some light cloud cover, whereas the POLDER reflectance has atmospheric effects
368   subtracted.  The effect of increasing Rayleigh scattering is seen for shorter wavelengths measured by
369   EPIC. The Khingan range is mainly covered by deciduous broadleaf and a mix of deciduous and
370   evergreen needle leaf forest with a small amount of grassland, while the area around Casper is mainly
371   short grass prairie land with few trees.

---

## Author Comment (AC3) · 16 May 2018

The paper has considerably improved. Most of the my comments and recommendations have been already mentioned by reviewer #1. From my point of view the Maignan et al., 2004 study has to be reported in the introduction sections and the "differences" observed with this study have to be reported in more detail in the section that currently this work is mentioned.

I think this is an interesting and unique work that should be published in AMT.

My co-authors and I have made minor revisions (spelling, punctuation, and slightly improved wording). I added a reference for the gamma correction, corrected the reference list and figure caption list, interchanged the order between figure 10 and 11, removed a duplicate definition of $R_{EN}(\lambda)$, fixed the caption of Fig. 6a, changed the symbol for percent difference PD to PDF to avoid confusion with "Probability Distribution", corrected the caption of Fig. 8a, 8b, and 8c to (A+D), corrected a date on page 21 from 19 to 20, Fig 11A albedo was miscalculated by a factor of $\pi$ while Fig. 11B is unchanged, the references to equations on pare 23 was changed from 1 and 2 to 2 and 3, and brackets were introduced for $<R_{Casper}>$ and $<R_{Columbia}>$ for eqns. 2 and 3.

I have added a new figure and text describing the comparison between POLDER and EPIC related to the Maignan et al. 2004 paper. I have also added a mention of the Maignan et al. paper in the introduction.

Measured backscattered radiances of the entire sunlit Earth were obtained during the 21 August 2017 eclipse from EPIC (Earth Polychromatic Imaging Camera) on the DSCOVR (Deep Space Climate Observatory) satellite. EPIC obtains synoptic observations of the Earth from an orbit around the $L_1$ point (Lagrange 1) 1.5 million km from Earth (Herman et al., 2018). EPIC top of the atmosphere TOA albedo measurements, made at a backscatter angle of $172^O$, are in the enhanced reflectivity regime (hotspot angles). EPIC non-eclipse day TOA albedos are compared to the satellite instrument POLDER (POLarization and Directionality of the Earth's Reflectances) surface reflectivity measurements at $8^O$ (Maignan et al., 2004).

**3.3 Comparison of EPIC albedo with POLDER reflectance**

The TOA albedo measurements made by EPIC can be compared with reflectance measurements made by the POLDER satellite instrument near the hotspot backscatter direction ($172^O$) for the incident solar irradiance over nearly cloud-free scenes (Maignan et al., 2004). EPIC C/s can be converted to albedo using the calibration constants $K(\lambda)$, which already contains the factor $\pi$ (Fig. 11A). The average TOA albedo from EPIC was almost the same on 20 Aug. as on 23 Aug. For EPIC albedo data over grassland common to Casper, Wyoming compared to the POLDER measurements, the C/s data for each wavelength (see Fig.5 for 443 nm) can be converted to TOA albedo.

Measurements from the POLDER satellite over Khingan Range, China (117.55°E to131.56°E, 45.68°N to 53.56°N) show that the backscatter amount from the land surface increases with increasing wavelength (Maignan et al., 2004). The Khingan range is mainly covered by deciduous broadleaf and a mix of deciduous and evergreen needle leaf forest with a small amount of grassland, while the area around Casper is mainly short grass prairie land with few trees. Over Casper, WY (Fig. 11B), the wavelength dependence of the EPIC TOA albedo (551, 680, and 780 nm) at $172^O$ backscatter angle is similar to POLDER surface reflectance at $8^O$. The shape and magnitude differences are partially caused by the atmospheric component of the albedo that includes some light cloud cover, whereas the POLDER reflectance has atmospheric effects subtracted. The effect of increasing Rayleigh scattering is seen for shorter wavelengths measured by EPIC.

[Figure]

Fig. 11 A. The measured albedo at Casper Wyoming on 20 Aug (black curve) and 23 Aug (grey curve) compared to B the POLDER measured surface reflectance in the Khingan Range, China ( Maignan et al., 2004) corresponding to $8^O$ from overhead sun.

---

## Author Response (AR2)

Bringing up the vegetation red-edge reflectivity is a good point for discussion.

"However, I do have a request. One question was already raised in the quick review of the initial submission, and this point wasn't addressed properly. In my view, Rayleigh scattering cannot be the cause for the wavelength-dependence of the "PDF" averaged over the globe (line 331 of the manuscript). Rayleigh scattering causes a smoothing of the image, in the sense that some radiation from outside the shadow is scattered into the shadow region. But i don't see why this should reduce the loss. Certainly, the shadow itself is brighter, but outside the shadow it's darker and in first approximation the integrated reflectance would be the same.

In terms of conservation of photons, this is certainly correct

Also, if Rayleigh scattering was the cause, I would have expected a wavelength dependence at the lower wavelengths, not just at the 4 higher ones. Between 317 and 551nm the Rayleigh scattering cross section varies by a factor of 10 (4th power of the wavelength) - yet all reduction values are the same.

After rethinking the problem for the global integral, the percent change in the UV should be almost independent of wavelength. Rayleigh scattering is essentially a smooth fog with strong wavelength dependence..

Between 680 and 780nm, Rayleigh varies much less, yet the reflectance changes from 10 to 13%. As suggested before, this has probably nothing to do with Rayleigh scattering but with the vegetation edge. The stronger attenuation starts exactly at the wavelengths were the vegetation reflectance starts increasing, and since the shadow is over land, the longer wavelengths are attenuated disproportionally since the shaded area is much brighter than the ocean and hence a shadow causes larger reduction compared to lower wavelengths where the land is equally dark as the ocean. Could you please consider that, or convince me that I am wrong?"

If I understand your argument, it is that more light is reflected from land vegetation for wavelengths longer than about 700 nm. Some of the vegetation reflected light is scattered from land onto ocean surfaces where it is absorbed (low reflectivity). This would cause a larger reduction from the eclipse than at shorter wavelengths where the land reflectivity is low.

I'm not sure that I agree with this argument. The same type of reflection and scattering occurs on non-eclipse days as occurs on eclipse days. The fractional change caused by the eclipse is just proportional to Fo*d, where d is the fractional obscuration. However, there is another effect. Light scatters from brighter regions to dimmer regions reducing the percent change caused by the eclipse. Except within the region of totality, this effect is very small. This is opposite of the numbers in Table 3. Plus, the O2 A-band (13%) shows a larger number than 780 nm (12%).

When observing this eclipse from Casper, the overhead sky was mostly dark with a ring of whitish light near the horizon. This horizon light caused the percentage change in illumination to decrease at the center of totality. The horizontal view is limited to no more 56 km by the earth's curvature. The light band was about 5 degrees of elevation above the horizon, which means it covered much greater distances. The white light reaching me was caused by atmospheric scattering in the visible range. A similar effect must occur at 780 nm, except that the Rayleigh scattering from bright regions into darker regions is smaller.  If there are clouds in the brighter region, then some light will be scattered into the darker region decreasing the percent change. Light scattering from high reflectivity regions (land) to low reflectivity regions (oceans) will be lost, but in the same proportion as on non-eclipse days.  I think that it is likely that there is insufficient data to explain the wavelength dependence. This is where your 3-D model could be a big help.

**I have revised (green) the paragraph to read**

Percent difference PDF($\lambda_i$) calculations for $\lambda_i$ = 317.5, 325, 340, 388, 443, 551, 680, 688, 764, and 780 nm, based on Eqn. 1 are summarized in Table 3A, yielding PDF($\lambda_i$) = 9, 9, 9, 9, 9, 9, 10, 10, 13, and 12 % reductions in backscattered radiances in the direction of $L_1$, respectively for Casper with similar values for Columbia. The PDF(764 nm) within the strongly absorbing $O_2$ A-band is 13 % for Casper and 14% for Columbia, even though the reflected ICs(764nm) is much lower than the surrounding non-absorbed bands. The fact that adjacent absorbed and non-absorbed wavelengths give consistent PDF($\lambda_i$) suggests that most of the effect comes from clouds. Eclipse effects for the short UV wavelengths are affected by Rayleigh scattering and clouds, and not much by the relatively low UV surface reflectivity (about 4%). Eclipse effects on outgoing radiances for wavelengths longer than about 700 nm are increased by vegetation reflectivity, even where the amount of clear-sky penetrating radiances are small for the O2 688 and 764 nm channels. There is insufficient information to explain the small observed wavelength dependence in Table 3.